# Probabilistically Rewired Message-Passing Neural Networks

**Chendi Qian**[*]
Computer Science Department
RWTH Aachen University, Germany
`chendi.qian@log.rwth-aachen.de`

**Andrei Manolache**[*]
Computer Science Department
University of Stuttgart, Germany
Bitdefender, Romania
`andrei.manolache@ki.uni-stuttgart.de`

**Kareem Ahmed, Zhe Zeng & Guy Van den Broeck**
Computer Science Department
University of California, Los Angeles, USA

**Mathias Niepert**[†]
Computer Science Department
University of Stuttgart, Germany

**Christopher Morris**[†]
Computer Science Department
RWTH Aachen University, Germany

## Abstract

Message-passing graph neural networks (MPNNs) emerged as powerful tools for processing graph-structured input. However, they operate on a fixed input graph structure, ignoring potential noise and missing information. Furthermore, their local aggregation mechanism can lead to problems such as over-squashing and limited expressive power in capturing relevant graph structures. Existing solutions to these challenges have primarily relied on heuristic methods, often disregarding the underlying data distribution. Hence, devising principled approaches for learning to infer graph structures relevant to the given prediction task remains an open challenge. In this work, leveraging recent progress in exact and differentiable $k$-subset sampling, we devise probabilistically rewired MPNNs (PR-MPNNs), which learn to add relevant edges while omitting less beneficial ones. For the first time, our theoretical analysis explores how PR-MPNNs enhance expressive power, and we identify precise conditions under which they outperform purely randomized approaches. Empirically, we demonstrate that our approach effectively mitigates issues like over-squashing and under-reaching. In addition, on established real-world datasets, our method exhibits competitive or superior predictive performance compared to traditional MPNN models and recent graph transformer architectures.

## 1 Introduction

Graph-structured data is prevalent across various application domains, including fields like chemo- and bioinformatics (Barabasi & Oltvai, 2004; Jumper et al., 2021; Reiser et al., 2022), combinatorial optimization (Cappart et al., 2023), and social-network analysis (Easley et al., 2012), highlighting the need for machine learning techniques designed explicitly for graphs. In recent years, *message-passing graph neural networks* (MPNNs) (Kipf & Welling, 2017; Gilmer et al., 2017; Scarselli et al., 2008b; Veličković et al., 2018) have become the dominant approach in this area, showing promising performance in tasks such as predicting molecular properties (Klicpera et al., 2020; Jumper et al., 2021) or enhancing combinatorial solvers (Cappart et al., 2023).

However, MPNNs have a limitation due to their local aggregation mechanism. They focus on encoding local structures, severely limiting their expressive power (Morris et al., 2019; 2021; Xu et al., 2019). In addition, MPNNs struggle to capture global or long-range information, possibly leading to phenomena

---

[*]These authors contributed equally.
[†]Co-senior authorship.

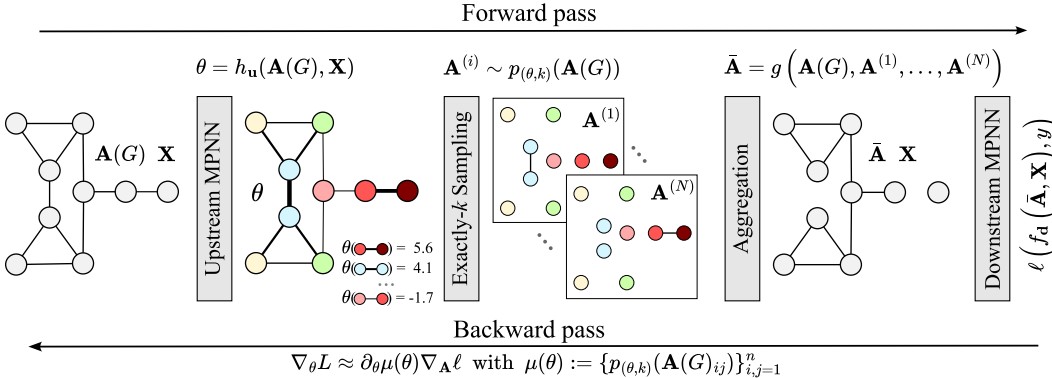

Figure 1: Overview of the probabilistically rewired MPNN framework. PR-MPNNs use an *upstream model* to learn priors $\boldsymbol{\theta}$ for candidate edges, parameterizing a probability mass function conditioned on exactly-$k$ constraints. Subsequently, we sample multiple $k$-edge adjacency matrices (here: $k = 1$) from this distribution, aggregate these matrices (here: subtraction), and use the resulting adjacency matrix as input to a *downstream model*, typically an MPNN, for the final predictions task. On the backward pass, the gradients of the loss $\ell$ regarding the parameters $\boldsymbol{\theta}$ are approximated through the derivative of the exactly-$k$ marginals in the direction of the gradients of the point-wise loss $\ell$ regarding the sampled adjacency matrix. We use recent work to make the computation of these marginals exact and differentiable, reducing both bias and variance.

like *under-reaching* (Barceló et al., 2020) or *over-squashing* (Alon & Yahav, 2021). Over-squashing, as explained by Alon & Yahav (2021), refers to excessive information compression from distant nodes due to a source node's extensive receptive field, occurring when too many layers are stacked.

Topping et al. (2021); Bober et al. (2022) investigated over-squashing from the perspective of Ricci and Forman curvature. Refining Topping et al. (2021), Di Giovanni et al. (2023) analyzed how the architectures' width and graph structure contribute to the over-squashing problem, showing that over-squashing happens among nodes with high commute time, stressing the importance of *graph rewiring techniques*, i.e., adding edges between distant nodes to make the exchange of information more accessible. In addition, Deac et al. (2022); Shirzad et al. (2023) utilized expander graphs to enhance message passing and connectivity, while Karhadkar et al. (2022) resort to spectral techniques, and Banerjee et al. (2022) proposed a greedy random edge flip approach to overcome over-squashing. Recent work (Gutteridge et al., 2023) aims to alleviate over-squashing by again resorting to graph rewiring. In addition, many studies have suggested different versions of multi-hop-neighbor-based message passing to maintain long-range dependencies (Abboud et al., 2022; Abu-El-Haija et al., 2019; Gasteiger et al., 2019; Xue et al., 2023), which can also be interpreted as a heuristic rewiring scheme. The above works indicate that graph rewiring is an effective strategy to mitigate over-squashing. However, most existing graph rewiring approaches rely on heuristic methods to add edges, potentially not adapting well to the specific data distribution or introducing edges randomly. Furthermore, there is limited understanding to what extent probabilistic rewiring, i.e., adding or removing edges based on the prediction task, impacts the expressive power of a model. In contrast to the above lines of work, graph transformers (Chen et al., 2022; Dwivedi et al., 2022b; He et al., 2023; Müller et al., 2023; Rampášek et al., 2022) and similar global attention mechanisms (Liu et al., 2021; Wu et al., 2021) marked a shift from local to global message passing, aggregating over all nodes. While not understood in a principled way, empirical studies indicate that graph transformers possibly alleviate over-squashing; see, e.g., Müller et al. (2023). However, due to their global aggregation mode, computing an attention matrix with $n^2$ entries for an $n$-order graph makes them applicable only to small or mid-sized graphs. Further, to capture non-trivial graph structure, they must resort to hand-engineered positional or structural encodings.

Overall, current strategies to mitigate over-squashing rely on heuristic rewiring methods that may not adapt well to a prediction task or employ computationally intensive global attention mechanisms. Furthermore, the impact of probabilistic rewiring on a model's expressive power remains unclear.

**Present work** By leveraging recent progress in differentiable $k$-subset sampling (Ahmed et al., 2023), we derive *probabilistically rewired MPNNs* (PR-MPNNs). Concretely, we utilize an *upstream model* to learn prior weights for candidate edges. We then utilize the weights to parameterize a probability distribution constrained by so-called $k$-subset constraints. Subsequently, we sample multiple $k$-edge adjacency matrices from this distribution and process them using a *downstream model*, typically an MPNN, for the final predictions task. To make this pipeline trainable via gradient descent, we adapt recently proposed discrete gradient estimation and tractable sampling techniques (Ahmed et al., 2023; Niepert et al., 2021;

Xie & Ermon, 2019); see Figure 1 for an overview of our architecture. Our theoretical analysis explores how PR-MPNNs overcome MPNNs' inherent limitations in expressive power and identifies precise conditions under which they outperform purely randomized approaches. Empirically, we demonstrate that our approach effectively mitigates issues like over-squashing and under-reaching. In addition, on established real-world datasets, our method exhibits competitive or superior predictive performance compared to traditional MPNN models and graph transformer architectures.

*Overall, PR-MPNNs pave the way for the principled design of more flexible MPNNs, making them less vulnerable to potential noise and missing information.*

## 1.1 RELATED WORK

MPNNs are inherently biased towards encoding local structures, limiting their expressive power (Morris et al., 2019; 2021; Xu et al., 2019). Specifically, they are at most as powerful as distinguishing non-isomorphic graphs or nodes with different structural roles as the 1-*dimensional Weisfeiler–Leman algorithm* (Weisfeiler & Leman, 1968), a simple heuristic for the graph isomorphism problem; see Section 2. Additionally, they cannot capture global or long-range information, often linked to phenomena such as under-reaching (Barceló et al., 2020) or over-squashing (Alon & Yahav, 2021), with the latter being heavily investigated in recent works.

**Graph rewiring** Several recent works aim to circumvent over-squashing via graph rewiring. Perhaps the most straightforward way of graph rewiring is incorporating multi-hop neighbors. For example, Brüel-Gabrielsson et al. (2022) rewires the graphs with $k$-hop neighbors and virtual nodes and also augments them with positional encodings. MixHop (Abu-El-Haija et al., 2019), SIGN (Frasca et al., 2020), DIGL (Gasteiger et al., 2019), and SP-MPNN (Abboud et al., 2022) can also be considered as graph rewiring as they can reach further-away neighbors in a single layer. Particularly, Gutteridge et al. (2023) rewires the graph similarly to Abboud et al. (2022) but with a novel delay mechanism, showcasing promising empirical results. Several rewiring methods depend on particular metrics, e.g., Ricci or Forman curvature (Bober et al., 2022) and balanced Forman curvature (Topping et al., 2021). In addition, Deac et al. (2022); Shirzad et al. (2023) utilize expander graphs to enhance message passing and connectivity, while Karhadkar et al. (2022) resort to spectral techniques, and Banerjee et al. (2022) propose a greedy random edge flip approach to overcome over-squashing. Refining Topping et al. (2021), Di Giovanni et al. (2023) analyzed how the architectures' width and graph structure contribute to the over-squashing problem, showing that over-squashing happens among nodes with high commute time, stressing the importance of rewiring techniques. Contrary to our proposed method, these strategies to mitigate over-squashing either rely on heuristic rewiring methods or use purely randomized approaches that may not adapt well to a given prediction task. Furthermore, the impact of existing rewiring methods on a model's expressive power remains unclear and we close this gap with our work.

There also exists a large set of works from the field of graph structure learning proposing heuristical graph rewiring approaches; see Appendix A for details.

**Graph transformers** Different from the above, graph transformers (Dwivedi et al., 2022b; He et al., 2023; Müller et al., 2023; Rampášek et al., 2022; Chen et al., 2022) and similar global attention mechanisms (Liu et al., 2021; Wu et al., 2021) marked a shift from local to global message passing, aggregating over all nodes. While not understood in a principled way, empirical studies indicate that graph transformers possibly alleviate over-squashing; see, e.g., Müller et al. (2023). However, all transformers suffer from their quadratic space and memory requirements due to computing an attention matrix.

## 2 BACKGROUND

In the following, we provide the necessary background.

**Notations** Let $\mathbb{N} := \{1, 2, 3, \dots\}$. For $n \geq 1$, let $[n] := \{1, \dots, n\} \subset \mathbb{N}$. We use $\{\!\{\dots\}\!\}$ to denote multisets, i.e., the generalization of sets allowing for multiple instances for each of its elements. A *graph* $G$ is a pair $(V(G), E(G))$ with *finite* sets of *vertices* or *nodes* $V(G)$ and *edges* $E(G) \subseteq \{\!\{u, v\} \subseteq V(G) \mid u \neq v\}$. If not otherwise stated, we set $n := |V(G)|$, and the graph is of *order* $n$. We also call the graph $G$ an $n$-order graph. For ease of notation, we denote the edge $\{u, v\}$ in $E(G)$ by $(u, v)$ or $(v, u)$. Throughout the paper, we use standard notations, e.g., we denote the *neighborhood* of a vertex $v$ by $N(v)$ and $\ell(v)$ denotes its discrete vertex label, and so on; see Appendix B for details.

**1-dimensional Weisfeiler–Leman algorithm** The 1-WL or color refinement is a well-studied heuristic for the graph isomorphism problem, originally proposed by Weisfeiler & Leman (1968). Formally, let $G = (V(G), E(G), \ell)$ be a labeled graph. In each iteration, $t > 0$, the 1-WL computes a node coloring $C_t^1 : V(G) \to \mathbb{N}$, depending on the coloring of the neighbors. That is, in iteration $t > 0$, we set

$$C_t^1(v) \coloneqq \mathsf{RELABEL}\Big(\big(C_{t-1}^1(v), \{\!\!\{ C_{t-1}^1(u) \mid u \in N(v) \}\!\!\}\big)\Big),$$

for all nodes $v \in V(G)$, where RELABEL injectively maps the above pair to a unique natural number, which has not been used in previous iterations. In iteration 0, the coloring $C_0^1 \coloneqq \ell$. To test if two graphs $G$ and $H$ are non-isomorphic, we run the above algorithm in "parallel" on both graphs. If the two graphs have a different number of nodes colored $c \in \mathbb{N}$ at some iteration, the 1-WL *distinguishes* the graphs as non-isomorphic. Moreover, if the number of colors between two iterations, $t$ and $(t+1)$, does not change, i.e., the cardinalities of the images of $C_t^1$ and $C_{i+t}^1$ are equal, or, equivalently,

$$C_t^1(v) = C_t^1(w) \iff C_{t+1}^1(v) = C_{t+1}^1(w),$$

for all nodes $v$ and $w$ in $V(G)$, the algorithm terminates. For such $t$, we define the *stable coloring* $C_\infty^1(v) = C_t^1(v)$, for $v$ in $V(G)$. The stable coloring is reached after at most $\max\{|V(G)|, |V(H)|\}$ iterations (Grohe, 2017). It is easy to see that the algorithm cannot distinguish all non-isomorphic graphs (Cai et al., 1992). Nonetheless, it is a powerful heuristic that can successfully test isomorphism for a broad class of graphs (Babai & Kucera, 1979). A function $f : V(G) \to \mathbb{R}^d$, for $d > 0$, is 1-*WL-equivalent* if $f \equiv C_\infty^1$; see Appendix B for details.

**Message-passing graph neural networks** Intuitively, MPNNs learn a vectorial representation, i.e., a $d$-dimensional real-valued vector, representing each vertex in a graph by aggregating information from neighboring vertices. Let $\mathbf{G} = (G, \mathbf{L})$ be an attributed graph, following, Gilmer et al. (2017) and Scarselli et al. (2008a), in each layer, $t > 0$, we compute vertex features

$$\mathbf{h}_v^{(t)} \coloneqq \mathsf{UPD}^{(t)}\Big(\mathbf{h}_v^{(t-1)}, \mathsf{AGG}^{(t)}\big(\{\!\!\{ \mathbf{h}_u^{(t-1)} \mid u \in N(v) \}\!\!\}\big)\Big) \in \mathbb{R}^d,$$

where $\mathsf{UPD}^{(t)}$ and $\mathsf{AGG}^{(t)}$ may be differentiable parameterized functions, e.g., neural networks, and $\mathbf{h}_v^{(t)} = \mathbf{L}_v$. In the case of graph-level tasks, e.g., graph classification, one uses

$$\mathbf{h}_G \coloneqq \mathsf{READOUT}\big(\{\!\!\{ \mathbf{h}_v^{(T)} \mid v \in V(G) \}\!\!\}\big) \in \mathbb{R}^d,$$

to compute a single vectorial representation based on learned vertex features after iteration $T$. Again, READOUT may be a differentiable parameterized function, e.g., a neural network. To adapt the parameters of the above three functions, they are optimized end-to-end, usually through a variant of stochastic gradient descent, e.g., Kingma & Ba (2015), together with the parameters of a neural network used for classification or regression.

## 3 PROBABILISTICALLY REWIRED MPNNS

Here, we outline probabilistically rewired MPNNs (PR-MPNNs) based on recent advancements in discrete gradient estimation and tractable sampling techniques (Ahmed et al., 2023). Let $\mathfrak{A}_n$ denote the set of adjacency matrices of $n$-order graphs. Further, let $(G, \mathbf{X})$ be a $n$-order attributed graph with an adjacency matrix $\mathbf{A}(G) \in \mathfrak{A}_n$ and node attribute matrix $\mathbf{X} \in \mathbb{R}^{n \times d}$, for $d > 0$. A PR-MPNN maintains a (parameterized) *upstream model* $h_{\boldsymbol{u}} : \mathfrak{A}_n \times \mathbb{R}^{n \times d} \to \Theta$, typically a neural network, parameterized by $\boldsymbol{u}$, mapping an adjacency matrix and corresponding node attributes to unnormalized edge priors $\boldsymbol{\theta} \in \Theta \subseteq \mathbb{R}^{n \times n}$.

In the following, we use the *priors* $\boldsymbol{\theta}$ as parameters of a (conditional) probability mass function,

$$p_{\boldsymbol{\theta}}(\mathbf{A}(H)) \coloneqq \prod_{i,j=1}^n p_{\theta_{ij}}(\mathbf{A}(H)_{ij}),$$

assigning a probability to each adjacency matrix in $\mathfrak{A}_n$, where $p_{\theta_{ij}}(\mathbf{A}(H)_{ij} = 1) = \mathrm{sigmoid}(\theta_{ij})$ and $p_{\theta_{ij}}(\mathbf{A}(H)_{ij} = 0) = 1 - \mathrm{sigmoid}(\theta_{ij})$. Since the parameters $\boldsymbol{\theta}$ depend on the input graph $G$, we can view the above probability as a conditional probability mass function conditioned on the graph $G$.

Unlike previous probabilistic rewiring approaches, e.g., Franceschi et al. (2019), we introduce dependencies between the graph's edges by conditioning the probability mass function $p_{\theta_{ij}}(\mathbf{A}(H))$ on a $k$-*subset constraint*. That is, the probability of sampling any given $k$-edge adjacency matrix $\mathbf{A}(H)$, becomes

$$p_{(\boldsymbol{\theta},k)}(\mathbf{A}(H)) := \begin{cases} p_{\boldsymbol{\theta}}(\mathbf{A}(H))/Z & \text{if } \|\mathbf{A}(H)\|_1 = k, \\ 0 & \text{otherwise}, \end{cases} \quad \text{with} \quad Z := \sum_{\mathbf{B} \in \mathfrak{A}_n \,:\, \|\mathbf{B}\|_1 = k} p_{\boldsymbol{\theta}}(\mathbf{B}). \quad (1)$$

The original graph $G$ is now rewired into a new adjacency matrix $\bar{\mathbf{A}}$ by combining $N$ samples $\mathbf{A}^{(i)} \sim p_{(\boldsymbol{\theta},k)}(\mathbf{A}(G))$ for $i \in [N]$ together with the original adjacency matrix $\mathbf{A}(G)$ using a differentiable aggregation function $g\colon \mathfrak{A}_n^{(N+1)} \to \mathfrak{A}_n$, i.e., $\bar{\mathbf{A}} := g(\mathbf{A}(G), \mathbf{A}^{(1)}, \dots, \mathbf{A}^{(N)}) \in \mathfrak{A}_n$. Subsequently, we use the resulting adjacency matrix as input to a *downstream model* $f_{\mathbf{d}}$, parameterized by $\mathbf{d}$, typically an MPNN, for the final predictions task.

We have so far assumed that the upstream MPNN computes one set of priors $h_{\boldsymbol{u}}\colon \mathfrak{A}_n \times \mathbb{R}^{n \times d} \to \mathbb{R}^{n \times n}$ which we use to generate a new adjacency matrix $\bar{\mathbf{A}}$ through sampling and then aggregating the adjacency matrices $\mathbf{A}^{(1)}, \dots, \mathbf{A}^{(N)}$. In Section 5, we show empirically that having multiple sets of priors from which we sample is beneficial. Multiple sets of priors mean that we learn an upstream model $h_{\boldsymbol{u}}\colon \mathfrak{A}_n \times \mathbb{R}^{n \times d} \to \mathbb{R}^{n \times n \times M}$ where $M$ is the number of priors. We can then sample and aggregate the adjacency matrices from these multiple sets of priors.

**Learning to sample** To learn the parameters of the up- and downstream model $\boldsymbol{\omega} = (\boldsymbol{u}, \boldsymbol{d})$ of the PR-MPNN architecture, we minimize the expected loss

$$L(\mathbf{A}(G), \mathbf{X}, y; \boldsymbol{\omega}) := \mathbb{E}_{\mathbf{A}^{(i)} \sim p_{(\boldsymbol{\theta},k)}(\mathbf{A}(G))}\Big[\ell\Big(f_{\boldsymbol{d}}\Big(g\Big(\mathbf{A}(G), \mathbf{A}^{(1)}, \dots, \mathbf{A}^{(N)}\Big), \mathbf{X}\Big), y\Big)\Big],$$

with $y \in \mathcal{Y}$, the targets, $\ell$ a point-wise loss such as the cross-entropy or MSE, and $\boldsymbol{\theta} = h_{\boldsymbol{u}}(\mathbf{A}(G), \mathbf{X})$. To minimize the above expectation using gradient descent and backpropagation, we need to efficiently draw Monte-Carlo samples from $p_{(\boldsymbol{\theta},k)}(\mathbf{A}(G))$ and estimate $\nabla_{\boldsymbol{\theta}} L$ the gradients of an expectation regarding the parameters $\boldsymbol{\theta}$ of the distribution $p_{(\boldsymbol{\theta},k)}$.

**Sampling** To sample an adjacency matrix $\mathbf{A}^{(i)}$ from $p_{(\boldsymbol{\theta},k)}(\mathbf{A}(G))$ conditioned on $k$-edge constraints, and to allow PR-MPNNs to be trained end-to-end, we use SIMPLE (Ahmed et al., 2023), a recently proposed gradient estimator. Concretely, we can use SIMPLE to sample *exactly* from the $k$-edge adjacency matrix distribution $p_{(\boldsymbol{\theta},k)}(\mathbf{A}(G))$ on the forward pass. On the backward pass, we compute the approximate gradients of the loss (which is an expectation over a discrete probability mass function) regarding the prior weights $\boldsymbol{\theta}$ using

$$\nabla_{\boldsymbol{\theta}} L \approx \partial_{\boldsymbol{\theta}} \mu(\boldsymbol{\theta}) \nabla_{\mathbf{A}} \ell \quad \text{with} \quad \mu(\boldsymbol{\theta}) := \{p_{(\boldsymbol{\theta},k)}(\mathbf{A}(G)_{ij})\}_{i,j=1}^n \in \mathbb{R}^{n \times n},$$

with an exact and efficient computation of the marginals $\mu(\boldsymbol{\theta})$ that is differentiable on the backward pass, achieving lower bias and variance. We show empirically that SIMPLE (Ahmed et al., 2023) outperforms other sampling and gradient approximation methods such as GUMBEL SOFTSUB-ST (Xie & Ermon, 2019) and I-MLE (Niepert et al., 2021), improving accuracy without incurring a computational overhead.

**Computational complexity** The vectorized complexity of the exact sampling and marginal inference step is $\mathcal{O}(\log k \log l)$, where $k$ is from our $k$-subset constraint, and $l$ is the maximum number of edges that we can sample. Assuming a constant number of layers, PR-MPNN's worst-case training complexity is $\mathcal{O}(l)$ for both the upstream and downstream models. Let $n$ be the number of nodes in the initial graph, and $l = \max(\{l_{\text{add}}, l_{\text{rm}}\})$, with $l_{\text{add}}$ and $l_{\text{rm}}$ being the maximum number of added and deleted edges. If we consider all of the possible edges for $l_{\text{add}}$, the worst-case complexity becomes $\mathcal{O}(n^2)$. Therefore, to reduce the complexity in practice, we select a subset of the possible edges using simple heuristics, such as considering the top $l_{\text{add}}$ edges of the most distant nodes. During inference, since we do not need gradients for edges not sampled in the forward pass, the complexity is $\mathcal{O}(l)$ for the upstream model and $\mathcal{O}(L)$ for the downstream model, with $L$ being the number of edges in the rewired graph.

## 4 EXPRESSIVE POWER OF PROBABILISTICALLY REWIRED MPNNS

We now, for the first time, explore the extent to which probabilistic MPNNs overcome the inherent limitations of MPNNs in expressive power caused by the equivalence to 1-WL in distinguishing non-isomorphic graphs (Xu et al., 2018; Morris et al., 2019). Moreover, we identify formal conditions under

which PR-MPNNs outperform popular randomized approaches such as those dropping nodes and edges uniformly at random. We first make precise what we mean by probabilistically separating graphs by introducing a probabilistic and generally applicable notion of graph separation.

Let us assume a conditional probability mass function $p \colon \mathfrak{A}_n \to [0,1]$ conditioned on a given $n$-order graph, defined over the set of adjacency matrices of $n$-order graphs. In the context of PR-MPNNs, $p$ is the probability mass function defined in Section 3 but it could also be any other conditional probability mass function over graphs. Moreover, let $f \colon \mathfrak{A}_n \to \mathbb{R}^d$, for $d > 0$, be a permutation-invariant, parameterized function mapping a sampled graph's adjacency matrix to a vector in $\mathbb{R}^d$. The function $f$ could be the composition of an aggregation function $g$ that removes the sampled edges from the input graph $G$ and of a downstream MPNN. Now, the conditional probability mass function $p$ *separates* two graphs $G$ and $H$ with probability $\rho$ *with respect to* $f$ if

$$\mathbb{E}_{\bar{G} \sim p(\cdot|G), \bar{H} \sim p(\cdot|H)} \big[ f(\mathbf{A}(\bar{G})) \neq f(\mathbf{A}(\bar{H})) \big] = \rho,$$

that is, if in expectation over the conditional probability distribution, the vectors $f(\mathbf{A}(\bar{G}))$ and $f(\mathbf{A}(\bar{H}))$ are distinct with probability $\rho$.

In what follows, we analyze the case of $p$ being the exactly-$k$ probability distribution defined in Equation 1 and $f$ being the aggregation function removing edges and a downstream MPNN. However, our framework readily generalizes to the case of node removal, and we provide these theoretical results in the appendix. Following Section 3, we sample adjacency matrices with exactly $k$ edges and use them to remove edges from the original graph. We aim to understand the separation properties of the probability mass function $p_{(k,\boldsymbol{\theta})}$ in this setting and for various types of graph structures. Most obviously, we do not want to separate isomorphic graphs and, therefore, remain isomorphism invariant, a desirable property of MPNNs.

**Theorem 4.1.** *For sufficiently large $n$, for every $\varepsilon \in (0,1)$ and $k > 0$, we have that for almost all pairs, in the sense of Babai et al. (1980), of isomorphic $n$-order graphs $G$ and $H$ and all permutation-invariant, $1$-WL-equivalent functions $f \colon \mathfrak{A}_n \to \mathbb{R}^d$, $d > 0$, there exists a probability mass function $p_{(\boldsymbol{\theta},k)}$ that separates the graph $G$ and $H$ with probability at most $\varepsilon$ with respect to $f$.*

Theorem 4.1 relies on the fact that most graphs have a discrete $1$-WL coloring. For graphs where the $1$-WL stable coloring consists of a discrete and non-discrete part, the following result shows that there exist distributions $p_{(\boldsymbol{\theta},k)}$ not separating the graphs based on the partial isomorphism corresponding to the discrete coloring.

**Proposition 4.2.** *Let $\varepsilon \in (0,1)$, $k > 0$, and let $G$ and $H$ be graphs with identical $1$-WL stable colorings. Let $V_G$ and $V_H$ be the subset of nodes of $G$ and $H$ that are in color classes of cardinality $1$. Then, for all choices of $1$-WL-equivalent functions $f$, there exists a conditional probability distribution $p_{(\boldsymbol{\theta},k)}$ that separates the graphs $G[V_G]$ and $H[V_H]$ with probability at most $\varepsilon$ with respect to $f$.*

Existing methods such as DropGNN (Papp et al., 2021) or DropEdge (Rong et al., 2020) are more likely to separate two (partially) isomorphic graphs by removing different nodes or edges between discrete color classes, i.e., on their (partially) isomorphic subgraphs. For instance, in the appendix, we prove that pairs of graphs with $m$ edges exist where the probability of non-separation under uniform edge sampling is at most $1/m$. This is undesirable as it breaks the MPNNs' permutation-invariance in these parts.

Now that we have established that distributions with priors from upstream MPNNs are more likely to preserve (partial) isomorphism between graphs, we turn to analyze their behavior in separating the non-discrete parts of the coloring. The following theorem shows that PR-MPNNs are more likely to separate non-isomorphic graphs than probability mass functions that remove edges or nodes uniformly at random.

**Theorem 4.3.** *For every $\varepsilon \in (0,1)$ and every $k > 0$, there exists a pair of non-isomorphic graphs $G$ and $H$ with identical and non-discrete $1$-WL stable colorings such that for every $1$-WL-equivalent function $f$,*

*(1) there exists a probability mass function $p_{(k,\boldsymbol{\theta})}$ that separates $G$ and $H$ with probability at least $(1 - \varepsilon)$ with respect to $f$;*

*(2) removing edges uniformly at random separates $G$ and $H$ with probability at most $\varepsilon$ with respect to $f$.*

Finally, we can also show a negative result, namely that there exist classes of graphs for which PR-MPNNs cannot do better than random sampling.

Table 1: Comparison between PR-MPNN and baselines on three molecular property prediction datasets. We report results for PR-MPNN with different gradient estimators for $k$-subset sampling: GUMBEL SOFTSUB-ST (Maddison et al., 2017; Jang et al., 2017; Xie & Ermon, 2019), I-MLE (Niepert et al., 2021), and SIMPLE (Ahmed et al., 2023) and compare them with the base downstream model, and two graph transformer architectures. The variant using SIMPLE consistently outperforms the base models and is competitive or better than the two graph transformers. We use **green** for the best model, **blue** for the second-best, and **red** for third. We note with + EDGE the instances where edge features are provided and with - EDGE when they are not.

| | | ZINC | | OGBG-MOLHIV | ALCHEMY |
| | | - EDGE ↓ | + EDGE ↓ | + EDGE ↑ | + EDGE ↓ |
|---|---|---|---|---|---|
| GIN BACKBONE | K-ST SAT | 0.166±0.007 | 0.115±0.005 | 0.625±0.039 | N/A |
| | K-SG SAT | 0.162±0.013 | **0.095**±0.002 | 0.613±0.010 | N/A |
| | BASE | 0.258±0.006 | 0.207±0.006 | **0.775**±0.011 | 11.12±0.690 |
| | BASE W. PE | 0.162±0.001 | 0.101±0.004 | 0.764±0.018 | 7.197±0.094 |
| | PR-MPNN_GMB (OURS) | 0.153±0.003 | 0.103±0.008 | 0.760±0.025 | **6.858**±0.090 |
| | PR-MPNN_IMLE (OURS) | **0.151**±0.001 | 0.104±0.008 | **0.774**±0.015 | **6.692**±0.061 |
| | PR-MPNN_SIM (OURS) | **0.139**±0.001 | **0.085**±0.002 | **0.795**±0.009 | **6.447**±0.057 |
| PNA | GPS | N/A | **0.070**±0.004 | **0.788**±0.010 | N/A |
| | K-ST SAT | 0.164±0.007 | 0.102±0.005 | 0.625±0.039 | N/A |
| | K-SG SAT | **0.131**±0.002 | **0.094**±0.008 | 0.613±0.010 | N/A |

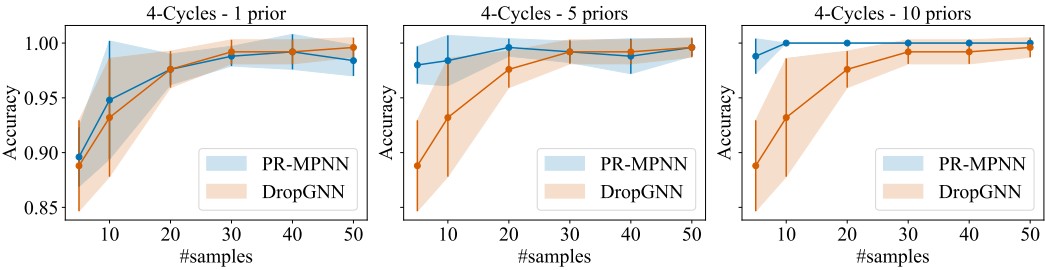

Figure 2: Comparison between PR-MPNN and DropGNN on the 4-CYCLES dataset. PR-MPNN rewiring is almost always better than randomly dropping nodes, and is always better with 10 priors.

**Proposition 4.4.** *For every $k > 0$, there exist non-isomorphic graphs $G$ and $H$ with identical 1-WL colorings such that every probability mass function $p_{(\theta,k)}$ separates the two graphs with the same probability as the distribution that samples edges uniformly at random.*

## 5 EXPERIMENTAL EVALUATION

Here, we explore to what extent our probabilistic graph rewiring leads to improved predictive performance on synthetic and real-world datasets. Concretely, we answer the following questions.

**Q1** Can probabilistic graph rewiring mitigate the problems of over-squashing and under-reaching in synthetic datasets?

**Q2** Is the expressive power of standard MPNNs enhanced through probabilistic graph rewiring? That is, can we verify empirically that the separating probability mass function of Section 4 can be learned with PR-MPNNs and that multiple priors help?

**Q3** Does the increase in predictive performance due to probabilistic rewiring apply to (a) graph-level molecular prediction tasks and (b) node-level prediction tasks involving heterophilic data?

An anonymized repository of our code can be accessed at `https://anonymous.4open.science/r/PR-MPNN`.

**Datasets** To answer **Q1**, we utilized the TREES-NEIGHBORSMATCH dataset (Alon & Yahav, 2021). Additionally, we created the TREES-LEAFCOUNT dataset to investigate whether our method could mitigate under-reaching issues; see Appendix E for details. To tackle **Q2**, we performed experiments with the EXP (Abboud et al., 2020) and CSL datasets (Murphy et al., 2019) to assess how much probabilistic graph rewiring can enhance the models' expressivity. In addition, we utilized the 4-CYCLES dataset from Loukas (2020); Papp et al. (2021) and set it against a standard DropGNN model (Papp et al.,

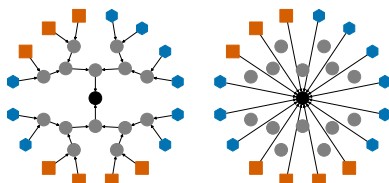

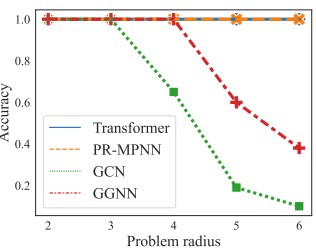

Figure 3: Example graph from the TREES-LEAFCOUNT test dataset with radius 4 (left). PR-MPNN rewires the graph, allowing the downstream MPNN to obtain the label information from the leaves in one massage-passing step (right).

Figure 4: Test accuracy of our rewiring method on the TREES-NEIGHBORSMATCH (Alon & Yahav, 2021) dataset, compared to the reported accuracies from Müller et al. (2023).

2021) for comparison while also ablating the performance difference concerning the number of priors and samples per prior. To answer **Q3** (a), we used the established molecular graph-level regression datasets ALCHEMY (Chen et al., 2019), ZINC (Jin et al., 2017; Dwivedi et al., 2020), OGBG-MOLHIV (Hu et al., 2020a), QM9 (Hamilton et al., 2017), LRGB (Dwivedi et al., 2022b) and five datasets from the TUDATASET repository (Morris et al., 2020). To answer **Q3** (b), we used the CORNELL, WISCONSIN, TEXAS node-level classification datasets (Pei et al., 2020).

**Baseline and model configurations** For our upstream model $h_{\boldsymbol{u}}$, we use an MPNN, specifically the GIN layer (Xu et al., 2019). For an edge $(v, w) \in E(G)$, we compute $\boldsymbol{\theta}_{vw} = \phi([\mathbf{h}_v^T || \mathbf{h}_w^T]) \in \mathbb{R}$, where $[\cdot || \cdot]$ is the concatenation operator and $\phi$ is an MLP. After obtaining the prior $\boldsymbol{\theta}$, we rewire our graphs by sampling two adjacency matrices for deleting edges and adding new edges, i.e., $g(\mathbf{A}(G), \mathbf{A}^{(1)}, \mathbf{A}^{(2)}) :=$ $(\mathbf{A}(G) - \mathbf{A}^{(1)}) + \mathbf{A}^{(2)}$ where $\mathbf{A}^{(1)}$ and $\mathbf{A}^{(2)}$ are two sampled adjacency matrices with a possibly different number of edges, respectively. Finally, the rewired adjacency matrix (or multiple adjacency matrices) is used in a *downstream model* $f_{\boldsymbol{d}} \colon \mathfrak{A}_n \times \mathbb{R}^{n \times d} \to \mathcal{Y}$, typically an MPNN, with parameters $\boldsymbol{d}$ and $\mathcal{Y}$ the prediction target set. For the instance where we have multiple priors, as described in Section 3, we can either aggregate the sampled adjacency matrices $\mathbf{A}^{(1)}, \ldots, \mathbf{A}^{(N)}$ into a single adjacency matrix $\bar{\mathbf{A}}$ that we send to a downstream model as described in Figure 1, or construct a downstream ensemble with multiple aggregated matrices $\bar{\mathbf{A}}_1, \ldots, \bar{\mathbf{A}}_M$. In practice, we always use a downstream ensemble before the final projection layer when we rewire with more than one adjacency matrix, and we do rewiring by both adding and deleting edges, please consult Table 8 in the Appendix for more details.

All of our downstream models $f_{\mathbf{d}}$ and base models are MPNNs with GIN layers. When we have access to edge features, we use the GINE variant (Hu et al., 2020b) for edge feature processing. For graph-level tasks, we use mean pooling, while for node-level tasks, we take the node embedding $\mathbf{h}_v^T$ for a node $v$. The final embeddings are then processed and projected to the target space by an MLP.

For ZINC, ALCHEMY, and OGBG-MOLHIV, we compare our rewiring approaches with the base downstream model, both with and without positional embeddings. Further, we compare to GPS (Rampášek et al., 2022) and SAT (Chen et al., 2022), two state-of-the-art graph transformers. For the TUDATASET, we compare with the reported scores from Giusti et al. (2023b) and use the same evaluation strategy as in Xu et al. (2019); Giusti et al. (2023b), i.e., running 10-fold cross-validation and reporting the maximum average validation accuracy. For different tasks, we search for the best hyperparameters for sampling and our upstream and downstream models. See Table 8 in the appendix for the complete description. For ZINC, ALCHEMY, and OGBG-MOLHIV, we evaluate multiple gradient estimators in terms of predictive power and computation time. Specifically, we compare GUMBEL SOFTSUB-ST (Maddison et al., 2017; Jang et al., 2017; Xie & Ermon, 2019), I-MLE (Niepert et al., 2021), and SIMPLE (Ahmed et al., 2023). The results in terms of predictive power are detailed in Table 1, and the computation time comparisons can be found in Table 9 in the appendix. Further experimental results on QM9 and LRGB are included in Appendix G in the appendix.

**Experimental results and discussion** Concerning **Q1**, our rewiring method achieves perfect test accuracy up to a problem radius of 6 on both TREES-NEIGHBORSMATCH and TREES-LEAFCOUNT, demonstrating that it can successfully alleviate over-squashing and under-reaching, see Figure 4. For TREES-LEAFCOUNT, our model can create connections directly from the leaves to the root, achieving perfect accuracy with a downstream model containing a single MPNN layer. We provide a qualitative result in Figure 3 and a detailed discussion in Appendix E. Concerning **Q2**, on the 4-CYCLES dataset, our probabilistic rewiring method matches or outperforms DropGNN. This advantage is most pronounced with 5 and 10 priors, where we achieve 100% task accuracy using 20 samples, as detailed in Figure 2. On the EXP dataset, we showcase the expressive power of probabilistic rewiring by achieving perfect

Table 2: Comparison between the base GIN model, PR-MPNN, and other more expressive models on the EXP dataset.

| MODEL | ACCURACY ↑ |
|---|---|
| GIN | $0.511_{\pm 0.021}$ |
| GIN + ID-GNN | $1.000_{\pm 0.000}$ |
| OSAN | $1.000_{\pm 0.000}$ |
| PR-MPNN (OURS) | $1.000_{\pm 0.000}$ |

Table 3: Comparison between the base GIN model and probabilistic rewiring model on CSL dataset, w/o positional encodings.

| MODEL | ACCURACY ↑ |
|---|---|
| GIN | $0.100_{\pm 0.000}$ |
| GIN + POSENC | $1.000_{\pm 0.000}$ |
| PR-MPNN (OURS) | $0.998_{\pm 0.008}$ |
| PR-MPNN + POSENC (OURS) | $1.000_{\pm 0.000}$ |

Table 4: Comparison between PR-MPNN and other approaches as reported in Giusti et al. (2023b); Karhadkar et al. (2022); Papp et al. (2021). Our model outperforms existing approaches while keeping a lower variance in most of the cases, except for NCI1, where the WL Kernel is the best. We use **green** for the best model, **blue** for the second-best, and **red** for third.

| MODEL | MUTAG | PTC_MR | PROTEINS | NCI1 | NCI109 |
|---|---|---|---|---|---|
| GK ($k = 3$) (SHERVASHIDZE ET AL., 2009) | $81.4_{\pm 1.7}$ | $55.7_{\pm 0.5}$ | $71.4_{\pm 0.3}$ | $62.5_{\pm 0.3}$ | $62.4_{\pm 0.3}$ |
| PK (NEUMANN ET AL., 2016) | $76.0_{\pm 2.7}$ | $59.5_{\pm 2.4}$ | $73.7_{\pm 0.7}$ | $82.5_{\pm 0.5}$ | N/A |
| WL KERNEL (SHERVASHIDZE ET AL., 2011) | $90.4_{\pm 5.7}$ | $59.9_{\pm 4.3}$ | $75.0_{\pm 3.1}$ | $\textbf{86.0}_{\pm 1.8}$ | N/A |
| DGCNN (ZHANG ET AL., 2018) | $85.8_{\pm 1.8}$ | $58.6_{\pm 2.5}$ | $75.5_{\pm 0.9}$ | $74.4_{\pm 0.5}$ | N/A |
| IGN (MARON ET AL., 2019B) | $83.9_{\pm 13.0}$ | $58.5_{\pm 6.9}$ | $76.6_{\pm 5.5}$ | $74.3_{\pm 2.7}$ | $72.8_{\pm 1.5}$ |
| GIN (XU ET AL., 2019) | $89.4_{\pm 5.6}$ | $64.6_{\pm 7.0}$ | $76.2_{\pm 2.8}$ | $82.7_{\pm 1.7}$ | N/A |
| PPGNS (MARON ET AL., 2019A) | $90.6_{\pm 8.7}$ | $66.2_{\pm 6.6}$ | $77.2_{\pm 4.7}$ | $83.2_{\pm 1.1}$ | $82.2_{\pm 1.4}$ |
| NATURAL GN (DE HAAN ET AL., 2020) | $89.4_{\pm 1.6}$ | $66.8_{\pm 1.7}$ | $71.7_{\pm 1.0}$ | $82.4_{\pm 1.3}$ | $83.0_{\pm 1.9}$ |
| GSN (BOURITSAS ET AL., 2022) | $92.2_{\pm 7.5}$ | $68.2_{\pm 7.2}$ | $76.6_{\pm 5.0}$ | $83.5_{\pm 2.0}$ | $83.5_{\pm 2.3}$ |
| CIN (BODNAR ET AL., 2021) | $92.7_{\pm 6.1}$ | $68.2_{\pm 5.6}$ | $77.0_{\pm 4.3}$ | $83.6_{\pm 1.4}$ | $\textcolor{red}{84.0}_{\pm 1.6}$ |
| CAN (GIUSTI ET AL., 2023A) | $\textcolor{red}{94.1}_{\pm 4.8}$ | $\textcolor{green}{72.8}_{\pm 8.3}$ | $\textcolor{blue}{78.2}_{\pm 2.0}$ | $84.5_{\pm 1.6}$ | $83.6_{\pm 1.2}$ |
| CIN++ (GIUSTI ET AL., 2023B) | $\textcolor{blue}{94.4}_{\pm 3.7}$ | $\textcolor{red}{73.2}_{\pm 6.4}$ | $\textcolor{green}{80.5}_{\pm 3.9}$ | $\textcolor{blue}{85.3}_{\pm 1.2}$ | $\textcolor{blue}{84.5}_{\pm 2.4}$ |
| FOSR (KARHADKAR ET AL., 2022) | $86.2_{\pm 1.5}$ | $58.5_{\pm 1.7}$ | $75.1_{\pm 0.8}$ | $72.9_{\pm 0.6}$ | $71.1_{\pm 0.6}$ |
| DROPGNN (PAPP ET AL., 2021) | $90.4_{\pm 7.0}$ | $66.3_{\pm 8.6}$ | $76.3_{\pm 6.1}$ | $81.6_{\pm 1.8}$ | $80.8_{\pm 2.6}$ |
| PR-MPNN (10-FOLD CV) | $\textcolor{green}{98.4}_{\pm 2.4}$ | $\textcolor{blue}{74.3}_{\pm 3.9}$ | $\textcolor{red}{80.7}_{\pm 3.9}$ | $\textcolor{green}{85.6}_{\pm 0.8}$ | $\textcolor{green}{84.6}_{\pm 1.2}$ |

accuracy, see Table 2. Besides, our rewiring approach can distinguish the regular graphs from the CSL dataset without any positional encodings, whereas the 1-WL-equivalent GIN obtains only random accuracy. Concerning **Q3** (a), the results in Table 1 show that our rewiring methods consistently outperform the base models on ZINC, ALCHEMY, and OGBG-MOLHIV and are competitive or better than the state-of-the-art GPS and SAT graph transformer methods. On TUDATASET, see Table 4, our probabilistic rewiring method outperforms existing approaches and obtains lower variance on most of the datasets, with the exception being NCI1, where our method ranks second, after the WL kernel. Hence, our results indicate that probabilistic graph rewiring can improve performance for molecular prediction tasks. Concerning **Q3** (b), we obtain performance gains over the base model and other existing MPNNs, see Table 11 in the appendix, indicating that data-driven rewiring has the potential of alleviating the *effects* of over-smoothing by removing undesirable edges and making new ones between nodes with similar features. The graph transformer methods outperform the rewiring approach and the base models, except on the TEXAS dataset, where our method gets the best result. We speculate that GIN's aggregation mechanism for the downstream models is a limiting factor on heterophilic data. We leave the analysis of combining probabilistic graph rewiring with downstream models that address over-smoothing for future investigations.

## 6 CONCLUSION

Here, we utilized recent advances in differentiable $k$-subset sampling to devise probabilistically rewired message-passing neural networks, which learn to add relevant edges while omitting less beneficial ones, resulting in the PR-MPNN framework. For the first time, our theoretical analysis explored how PR-MPNNs enhance expressive power, and we identified precise conditions under which they outperform purely randomized approaches. On synthetic datasets, we demonstrated that our approach effectively alleviates the issues of over-squashing and under-reaching while overcoming MPNNs' limits in expressive power. In addition, on established real-world datasets, we showed that our method is competitive or superior to conventional MPNN models and graph transformer architectures regarding predictive performance and computational efficiency. Ultimately, PR-MPNNs represent a significant step towards systematically developing more adaptable MPNNs, rendering them less susceptible to potential noise and missing data, thereby enhancing their applicability and robustness.

ACKNOWLEDGMENTS

CQ and CM are partially funded by a DFG Emmy Noether grant (468502433) and RWTH Junior Principal Investigator Fellowship under Germany's Excellence Strategy. AM and MN acknowledge funding by Deutsche Forschungsgemeinschaft (DFG, German Research Foundation) under Germany's Excellence Strategy - EXC 2075 – 390740016, the support by the Stuttgart Center for Simulation Science (SimTech), and the International Max Planck Research School for Intelligent Systems (IMPRS-IS). This work was funded in part by the DARPA Perceptually-enabled Task Guidance (PTG) Program under contract number HR00112220005, the DARPA Assured Neuro Symbolic Learning and Reasoning (ANSR) Program, and a gift from RelationalAI. GVdB discloses a financial interest in RelationalAI.

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

## A    ADDITIONAL RELATED WORK

In the following, we discuss additional related work.

**Graph structure learning** The field of graph structure learning (GSL) is a topic related to graph rewiring. Motivated by robustness and more general purposes, several GSL works have been proposed. Jin et al. (2020) optimizes a graph structure from scratch with some loss function as bias. More generally, an edge scorer function is learned, and modifications are made to the original graph structure (Chen et al., 2020; Yu et al., 2021; Zhao et al., 2021). To introduce discreteness and sparsity, Kazi et al. (2022); Franceschi et al. (2019); Zhao et al. (2021) leverage Gumbel and Bernoulli discrete sampling, respectively. Saha et al. (2023) incorporates end-to-end differentiable discrete sampling through the smoothed-Heaviside function. Moreover, GSL also benefits from self-supervised or unsupervised learning approaches; see, e.g., Zou et al. (2023); Fatemi et al. (2021); Liu et al. (2022a;b). For a comprehensive survey of GSL. see Fatemi et al. (2023); Zhou et al. (2023). In the context of node classification, there has been recent progress in understanding the interplay between graph structure and features (Castellana & Errica, 2023).

The main differences to the proposed PR-MPNN framework are as follows: (a) for sparsification, existing GSL approaches typically use a $k$-NN algorithm, a simple randomized version of $k$-NN, or model edges with independent Bernoulli random variables. In contrast, PR-MPNN uses a proper probability mass function derived from exactly-$k$ constraints. Hence, we introduce complex dependencies between the edge random variables and trade-off exploration and exploitation during training, and (b) GSL approaches do not use exact sampling of the exactly-$k$ distribution and recent sophisticated gradient estimation techniques. However, the theoretical insights we provide in this paper also largely translate to GSL approaches with the difference that sampling is replaced with an $\arg\max$ operation and, therefore, should be of independent interest to the GSL community.

## B    EXTENDED NOTATION

A *graph* $G$ is a pair $(V(G), E(G))$ with *finite* sets of *vertices* or *nodes* $V(G)$ and *edges* $E(G) \subseteq \{\{u, v\} \subseteq V(G) \mid u \neq v\}$. If not otherwise stated, we set $n \coloneqq |V(G)|$, and the graph is of *order* $n$. We also call the graph $G$ an $n$-order graph. For ease of notation, we denote the edge $\{u, v\}$ in $E(G)$ by $(u, v)$ or $(v, u)$. A *(vertex-)labeled graph* $G$ is a triple $(V(G), E(G), \ell)$ with a (vertex-)label function $\ell \colon V(G) \to \mathbb{N}$. Then $\ell(v)$ is a *label* of $v$, for $v$ in $V(G)$. An *attributed graph* $G$ is a triple $(V(G), E(G), a)$ with a graph $(V(G), E(G))$ and (vertex-)attribute function $a \colon V(G) \to \mathbb{R}^{1 \times d}$, for

some $d > 0$. That is, contrary to labeled graphs, we allow for vertex annotations from an uncountable set. Then $a(v)$ is an *attribute* or *feature* of $v$, for $v$ in $V(G)$. Equivalently, we define an $n$-order attributed graph $G \coloneqq (V(G), E(G), a)$ as a pair $\mathbf{G} = (G, \mathbf{L})$, where $G = (V(G), E(G))$ and $\mathbf{L}$ in $\mathbb{R}^{n \times d}$ is a *node attribute matrix*. Here, we identify $V(G)$ with $[n]$. For a matrix $\mathbf{L}$ in $\mathbb{R}^{n \times d}$ and $v$ in $[n]$, we denote by $\mathbf{L}_{v.}$ in $\mathbb{R}^{1 \times d}$ the $v$th row of $\mathbf{L}$ such that $\mathbf{L}_{v.} \coloneqq a(v)$. Furthermore, we can encode an $n$-order graph $G$ via an *adjacency matrix* $\mathbf{A}(G) \in \{0, 1\}^{n \times n}$, where $A_{ij} = 1$ if, and only, if $(i, j) \in E(G)$. We also write $\mathbb{R}^d$ for $\mathbb{R}^{1 \times d}$.

The *neighborhood* of $v$ in $V(G)$ is denoted by $N(v) \coloneqq \{u \in V(G) \mid (v, u) \in E(G)\}$ and the *degree* of a vertex $v$ is $|N(v)|$. Two graphs $G$ and $H$ are *isomorphic* and we write $G \simeq H$ if there exists a bijection $\varphi \colon V(G) \to V(H)$ preserving the adjacency relation, i.e., $(u, v)$ is in $E(G)$ if and only if $(\varphi(u), \varphi(v))$ is in $E(H)$. Then $\varphi$ is an *isomorphism* between $G$ and $H$. In the case of labeled graphs, we additionally require that $l(v) = l(\varphi(v))$ for $v$ in $V(G)$, and similarly for attributed graphs. Further, we call the equivalence classes induced by $\simeq$ isomorphism types.

A *node coloring* is a function $c \colon V(G) \to \mathbb{R}^d$, $d > 0$, and we say that $c(v)$ is the *color* of $v \in V(G)$. A node coloring induces an *edge coloring* $e_c \colon E(G) \to \mathbb{N}$, where $(u, v) \mapsto \{c(u), c(v)\}$ for $(u, v) \in E(G)$. A node coloring (edge coloring) $c$ *refines* a node coloring (edge coloring) $d$, written $c \sqsubseteq d$ if $c(v) = c(w)$ implies $d(v) = d(w)$ for every $v, w \in V(G)$ ($v, w \in E(G)$). Two colorings are equivalent if $c \sqsubseteq d$ and $d \sqsubseteq c$, in which case we write $c \equiv d$. A *color class* $Q \subseteq V(G)$ of a node coloring $c$ is a maximal set of nodes with $c(v) = c(w)$ for every $v, w \in Q$. A node coloring is called *discrete* if all color classes have cardinality 1.

## C   MISSING PROOFS

In the following, we outline missing proofs from the main paper.

**Theorem C.1** (Theorem 4.1 in the main paper). *For sufficiently large $n$, for every $\varepsilon \in (0, 1)$ and $k > 0$, we have that for almost all pairs, in the sense of Babai et al. (1980), of isomorphic $n$-order graphs $G$ and $H$ and all permutation-invariant, 1-$\mathsf{WL}$-equivalent functions $f \colon \mathfrak{A}_n \to \mathbb{R}^d$, $d > 0$, there exists a conditional probability mass function $p_{(\boldsymbol{\theta}, k)}$ that separates the graph $G$ and $H$ with probability at most $\varepsilon$ regarding $f$.*

Before proving the above result, we first need three auxiliary results. The first one is the well-known universal approximation theorem for multi-layer perceptrons.

**Theorem C.2** (Cybenko (1992); Leshno et al. (1993)). *Let $\sigma \colon \mathbb{R} \to \mathbb{R}$ be continuous and not polynomial. Then for every continuous function $f \colon K \to \mathbb{R}^n$, where $K \subseteq \mathbb{R}^m$ is a compact set, and every $\varepsilon > 0$ there is a depth-two multi-layer perceptron $N$ with activation function $\sigma^{(1)} = \sigma$ on layer 1 and no activation function on layer 2 (i.e., $\sigma^{(2)}$ is the identity function) computing a function $f_N$ such that*

$$\sup_{\mathbf{x} \in K} \|f(\mathbf{x}) - f_N(\mathbf{x})\| < \varepsilon.$$

Building on the first, the second result shows that an MPNN can approximate real-valued node colorings of a given finite graph arbitrarily close.

**Lemma C.3.** *Let $G$ be an $n$-order graph and let $c \colon V(G) \to \mathbb{R}^d$, $d > 0$, be a 1-$\mathsf{WL}$-equivalent node coloring. Then, for all $\varepsilon > 0$, there exists a (permutation-equivariant) MPNN $f \colon V(G) \to \mathbb{R}^d$, such that*

$$\max_{v \in V(G)} \|f(v) - c(v)\| < \varepsilon.$$

*Proof sketch.* First, by (Morris et al., 2019, Theorem 2), there exists an 1-$\mathsf{WL}$-equivalent MPNN $m \colon V(G) \to \mathbb{R}^d$ such that

$$c \equiv m.$$

Since the graph's number of vertices, by assumption, is finite, the cardinality of the image $K \coloneqq m^{-1}$ is also finite. Hence, we can find a continuous function $g \colon K \to \mathbb{R}^d$ such that $(g \circ m)(v) = c(v)$ for $v \in V(G)$. Since $K$ is finite and hence compact and $g$ is continuous, by Theorem C.2, we can approximate it arbitrarily close with a two-layer multi-layer perceptron, implying the existence of the MPNN $f$. □

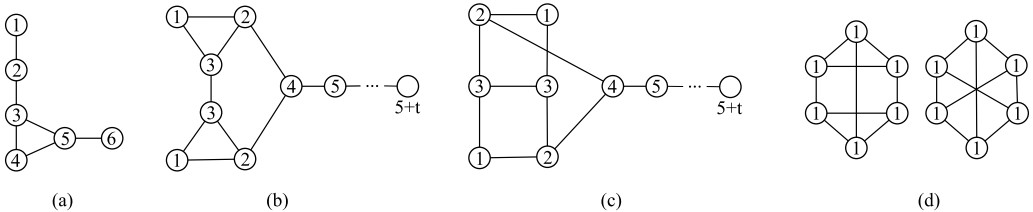

Figure 5: Example graphs used in the theoretical analysis.

The third result lifts the previous result to edge colorings.

**Lemma C.4.** *Let $G$ be an $n$-order graph and let $c\colon E(G) \to \mathbb{R}^d$, $d > 0$, be a 1-WL-equivalent edge coloring. Then, for all $\varepsilon > 0$, there exists a (permutation-equivariant) MPNN $f\colon E(G) \to \mathbb{R}^d$, such that*

$$\max_{e \in E(G)} \|f(e) - c(e)\| < \varepsilon.$$

*Proof sketch.* The proof is analogous to the proof of Lemma C.3. □

We note here that we can extend the above results to any finite subset of $n$-order graphs. We are now ready to prove Theorem C.1.

*Proof sketch.* Following Babai et al. (1980), for a sufficiently large order $n$, the 1-WL will compute a discrete coloring for almost any $n$-order graph. Concretely, they showed that an algorithm equivalent to the 1-WL computes a discrete coloring of graphs sampled from the Erdős–Rényi random graph model $G(n, 1/2)$ with the probability of failure bounded by $\mathcal{O}(n^{-1/7})$. Since the $G(n, 1/2)$ model assigns a uniform distribution over all graphs, the 1-WL succeeds on "almost all" graphs.

By the above, and due to Lemmas C.3 and C.4, every node, and thereby any edge, can be assigned a distinct arbitrary prior weight with an upstream MPNN. Consequently, there exists an upstream MPNN that returns a high prior $\boldsymbol{\theta}_{ij}$ for exactly $k$ edges ($\boldsymbol{\theta}_i$ for exactly $k$ nodes) such that sampling from the exactly-$k$ distribution returns these $k$ edges (nodes) with probability at least $\sqrt{1-\varepsilon}$. Specifically, we know that the upstream MPNN can return arbitrary priors $\boldsymbol{\theta}_{ij}$, and we want to show that, given some $\delta \in (0, 1)$, there exists at least one $\boldsymbol{\theta}$ such that for a set $S$ of $k$ edges (nodes) of $G$, we have $p_{\boldsymbol{\theta}, k}(S) \geq \delta$.

Let $m$ be the number of edges in the graph $G$ That is, from our probability definition, we obtain $p_{\boldsymbol{\theta}}(S) \geq \delta Z$. Without losing generality, let $w_1$ and $w_2$ be two prior weights with $w_1 > w_2$ such that $\theta_i = w_1$ for the edges (nodes) in $S$ and $\theta_i = w_2$ otherwise. Then $p_{\boldsymbol{\theta}}(S) \geq \delta Z$ becomes $w_1^k \geq \delta(\sum_{i=0}^{k} \binom{k}{i}\binom{m-k}{k-i} w_1^i w_2^{k-i})$. We use the upper bound $Z \leq w_1^k + (\binom{m}{k} - 1) w_2 w_1^{k-1}$ and obtain $w_2 \leq (1-\delta) w_1 \delta^{-1} (\binom{m}{k} - 1)^{-1}$. Therefore, a prior $\boldsymbol{\theta}$ exists, and we can obtain it by using the derived inequality. Now, we can set $\delta = \sqrt{1-\varepsilon}$. The sampled $k$ edges (nodes) are then identical for both graphs with probability at least $\sqrt{1-\varepsilon}^2 = 1 - \varepsilon$ and, therefore, the edges (nodes) that are removed are isomorphic edges (nodes) with probability at least $1 - \varepsilon$. When we remove pairs of edges (nodes) from two isomorphic graphs that are mapped to each other via an isomorphism, the graphs remain isomorphic and, therefore, must have the same 1-WL coloring. Since the two graphs have the same 1-WL coloring, an MPNN downstream model $f$ cannot separate them. □

**Proposition C.5** (Proposition 4.2 in the main paper). *Let $\varepsilon \in (0, 1)$, $k > 0$, and let $G$ and $H$ be graphs with identical 1-WL stable colorings. Let $V_G$ and $V_H$ be the subset of nodes of $G$ and $H$ that are in color classes of cardinality 1. Then, for all choices of 1-WL-equivalent functions $f$, there exists a conditional probability mass function $p_{(\boldsymbol{\theta}, k)}$ that separates the graphs $G[V_G]$ and $H[V_H]$ with probability at most $\varepsilon$ regarding $f$.*

*Proof sketch.* Since the graphs $G[V_G]$ and $H[V_H]$ have a discrete coloring under the 1-WL and the graphs $G$ and $H$ have identical 1-WL colorings, it follows that there exists an isomorphism $\varphi\colon G[V_G] \to H[V_H]$.

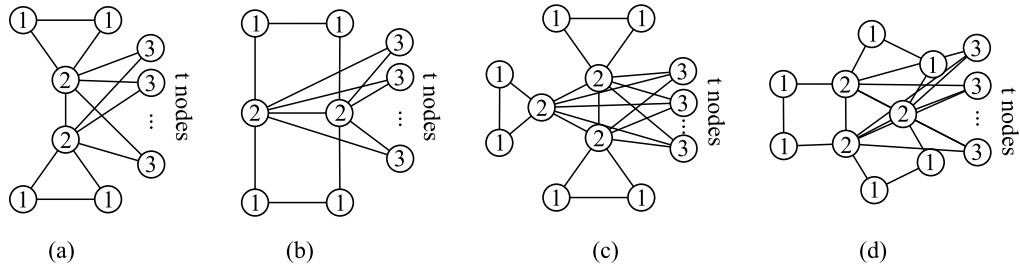

Figure 6: The graphs used in the proof of Theorem C.6 for node sampling.

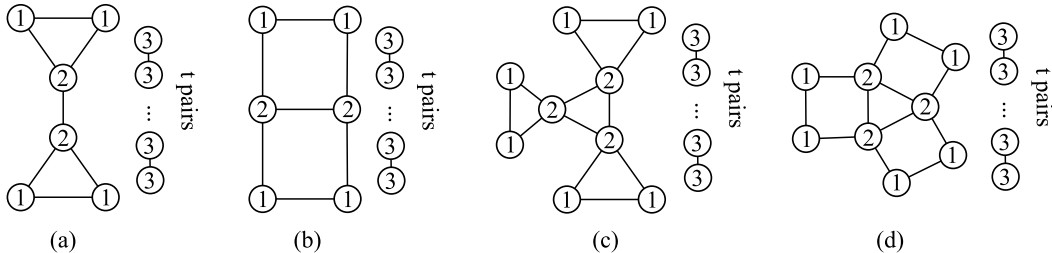

Figure 7: The graphs used in the proof of Theorem C.6 for edge sampling.

Analogous to the proof of Theorem 4.1, we can now show that there exists a set of prior weights that ensures that the exactly-$k$ sample selects the same subset of edges (nodes) from, respectively, the same subset of edges from $G_1[V_1]$ and $G_2[V_2]$ (the same subset of nodes from $V_G$ and $V_H$) with probability at least $\sqrt{1-\epsilon}$. Note that the cardinality of the sampled subsets could also be empty since the priors could be putting a higher weight on nodes (edges) with non-discrete color classes. □

Regarding uniform edge removal, consider the two graphs in Figure 5 (b) and (c). With a distribution based on an MPNN upstream model, the probability of separating the graphs by removing edges in the isomorphic subgraphs induced by the nodes with colors 4 to $5 + t$ can be made arbitrarily small for any $t$. However, the graphs would still be separated through samples in the parts whose coloring is non-discrete. In contrast, sampling uniformly at random separates the graphs in these isomorphic subgraphs with probability converging towards 1 with increasing $t$.

**Theorem C.6.** *For every $\varepsilon \in (0,1)$ and every $k > 0$, there exists a pair of non-isomorphic graphs $G$ and $H$ with identical and non-discrete 1-WL stable colorings such that for every 1-WL-equivalent function $f$*

*(1) there exists a probability mass function $p_{(k,\boldsymbol{\theta})}$ that separates $G$ and $H$ with probability at least $(1-\varepsilon)$ with respect to $f$;*

*(2) removing edges uniformly at random separates $G$ and $H$ with probability at most $\varepsilon$ with respect to $f$.*

*Proof.* We distinguish the two cases: (1) sampling nodes to be removed and (2) sampling edges to be removed from the original graphs.

For case (1), where we sample nodes to be removed, consider the graphs in Figure 6. For $k = 1$, we take the graphs (a) and (b). Both of these graphs have the same 1-WL coloring, indicated by the color numbers of the nodes. To separate the graphs, we need to sample and remove one of the nodes with color 1 or 2. Removing a node with color 3 would lead again to an identical color partition for the two graphs. Removing nodes with color 1 or 2 is achievable by placing a high prior weight $\boldsymbol{\theta}_u$ on nodes of *one* of the corresponding color classes; see Lemma C.3. Without loss of generality, we choose all nodes $u$ in color class 2 and set the prior weight $\boldsymbol{\theta}_u$ such that a node in this color class is sampled with probability $\sqrt{1-\varepsilon}$. Since a random sampler would uniformly sample any of the nodes, we simply have to increase the number $t$ of nodes with color 3 such that the probability of randomly sampling a node of the color classes 1 or 2 is smaller than or equal to $\sqrt{\epsilon}$.

For $k > 1$, we construct the graphs depicted in Figure 6 (c) and (d) for $k = 2$. These are constructed by first taking a $(k + 1)$-cycle and connecting to each node of the cycle the nodes with color class 1 in the two ways shown in Figure 6 (c) and (d). Finally, we connect $t$ nodes of color class 3 to each of the nodes in the cycle. These graphs can be separated by sampling $k$ nodes from either the color class 1 or 2. For instance, removing $k$ nodes from color class 2 always creates $k$ disconnected subgraphs of size 2 in the first parameterized graph but not the second. By Lemma C.3, we know that we can find an upstream model that leads to prior weights $\boldsymbol{\theta}_u$ such that sampling $k$ nodes from a color class has probability at least $\sqrt{1 - \epsilon}$; see the proof of Theorem C.1. As argued before, by increasing the number of nodes with color class 3, we can make the probability that a uniform sampler picks a node with color classes 1 or 2 to be less than or equal to $\sqrt{\epsilon}$.

For case (2), where we sample edges to be removed from the original graph, consider the graphs in Figure 7. For $k = 1$, we take the graphs (a) and (b). Both of these graphs have the same 1-WL coloring, indicated by the color numbers of the nodes. To separate the graphs, we need to sample from each graph an edge $(u, v)$ such that either $C_\infty^1(u) = C_\infty^1(v) = 1$ or $C_\infty^1(u) = 1$ and $C_\infty^1(v) = 2$. Removing an edge between two nodes with color class 3 in both graphs would lead again to an identical color partition of the two graphs. Removing an edge between the color classes $(1, 1)$ and $(1, 2)$ is possible by Lemma C.4 and choosing a prior weight large enough such that the probability of sampling an edge between these color classes is at least $\sqrt{1 - \epsilon}$; see the proof of Theorem C.1. Since a random sampler would sample an edge uniformly at random, we simply have to increase the number of nodes with color class 3 such that the probability of sampling an edge between the color classes $(1, 2)$ or $(1, 2)$ is smaller than $\sqrt{\epsilon}$.

For $k > 1$, we construct the graphs depicted in Figure 7(c) and (d) for $k = 2$. We first take a $(k+1)$-cycle and connect to each node of the said cycle the nodes with color classes 1 in the two different ways shown in Figure 7(c) and (d). Finally, we again add pairs of connected nodes with color class 3. The two graphs can be separated by sampling $k$ edges between the color classes $(1, 1)$, $(1, 2)$, and $(2, 2)$. For instance, sampling $k$ edges between nodes in color class 2 leads to a disconnected subgraph of size 3 in the first graph but not the second. By Lemma C.4, we know that we can learn an upstream MPNN that results in prior edge weights $\boldsymbol{\theta}_{uv}$ for all edges $(u, v)$ where both $u$ and $v$ are in color class 2, such that sampling $k$ of these edges has probability at least $\sqrt{1 - \epsilon}$; see the proof of Theorem C.1. Again, by increasing the number of nodes with color class 3, we can make the probability that a uniform sampler picks an edge between the color classes $(1, 1)$, $(1, 2)$ or $(2, 2)$ to be less than or equal to $\sqrt{\epsilon}$. $\qquad\square$

Finally, we can also show a negative result, i.e., graphs exist such that PR-MPNNs cannot do better than random sampling.

**Proposition C.7** (Proposition 4.4 in the main paper). *For every $k > 0$, there exist non-isomorphic graphs $H$ and $H$ with identical 1-WL colorings such that every probability mass function $p_{(\boldsymbol{\theta}, k)}$ separates the two graphs with the same probability as the distribution that samples nodes (edges) uniformly at random.*

*Proof.* Any pair of graphs where the 1-WL coloring consists of a single color class suffices to show the result. For instance, consider the graphs in Figure 5(d), where all nodes have the same color. In fact, any pair of non-isomorphic $d$-regular graphs for $d > 0$ works here. An MPNN upstream model cannot separate the prior weights of the nodes and, therefore, behaves as a uniform sampler. $\qquad\square$

# D   SIMPLE: SUBSET IMPLICIT LIKELIHOOD

In this section, we introduce SIMPLE, which is a main component of our work. The goal of SIMPLE is to build a gradient estimator for $\nabla_{\boldsymbol{\theta}} L(\mathbf{X}, y; \boldsymbol{\omega})$. It is inspired by a hypothetical sampling-free architecture, where the downstream neural network $f_{\mathbf{d}}$ is a function of the marginals, $\boldsymbol{\mu} := \mu(\boldsymbol{\theta}) := \{p_{\boldsymbol{\theta}}(z_j \mid \sum_i z_i = k)\}_{j=1}^n$, instead of a discrete sample $\boldsymbol{z}$, resulting in a loss $L_m$ s.t.

$$\nabla_{\boldsymbol{\theta}} L_m(\mathbf{X}, y; \boldsymbol{\omega}) = \partial_{\boldsymbol{\theta}} \mu(\boldsymbol{\theta})^\top \nabla_{\boldsymbol{\mu}} \ell_m(f_{\boldsymbol{d}}(\boldsymbol{\mu}, \mathbf{X}), y).$$

When the marginals $\mu(\boldsymbol{\theta})$ can be efficiently computed and differentiated, such a hypothetical pipeline can be trained end-to-end. Furthermore, Domke (2010) observed that, for an arbitrary loss function $\ell_m$ defined on the marginals, the Jacobian of the marginals w.r.t. the logits is symmetric. Consequently, computing the gradient of the loss w.r.t. the logits, $\nabla_{\boldsymbol{\theta}} L_m(\mathbf{X}, y; \boldsymbol{\omega})$, reduces to computing the *directional derivative*, or the Jacobian-vector product, of the marginals w.r.t. the logits in the direction of the gradient

of the loss. This offers an alluring opportunity, i.e., the conditional marginals characterize the probability of each $z_i$ in the sample, and could be thought of as a differentiable proxy for the samples. Specifically, by reparameterizing $\boldsymbol{z}$ as a function of the conditional marginal $\boldsymbol{\mu}$ under approximation $\partial_{\boldsymbol{\mu}} \boldsymbol{z} \approx \mathbf{I}$ as proposed by Niepert et al. (2021), and using the straight-through estimator for the gradient of the sample w.r.t. the marginals on the backward pass, SIMPLE approximate

$$\nabla_{\boldsymbol{\theta}} L(\mathbf{X}, y; \boldsymbol{\omega}) \approx \partial_{\boldsymbol{\theta}} \mu(\boldsymbol{\theta}) \nabla_{\boldsymbol{z}} L(\mathbf{X}, y; \boldsymbol{\omega}),$$

where the directional derivative of the marginals can be taken along *any downstream gradient*, rendering the whole pipeline end-to-end learnable despite the presence of sampling.

Now, estimating the gradient of the loss w.r.t. the parameters can be thought of as decomposing into two sub-problems: **(P1)** Computing the derivatives of conditional marginals $\partial_{\boldsymbol{\theta}} \mu(\boldsymbol{\theta})$, which requires the computation of the conditional marginals, and **(P2)** Computing the gradient of the loss w.r.t. the samples $\nabla_{\boldsymbol{z}} L(\mathbf{X}, y; \boldsymbol{\omega})$ using sample-wise loss, which requires drawing exact samples. These two problems are complicated by conditioning on the $k$-subset constraint, which introduces intricate dependencies to the distribution and is infeasible to solve naively, e.g., by enumeration. Next, we will show the solutions that SIMPLE provide to each problem, at the heart of which is the insight that we need not care about the variables' order, only their sum, introducing symmetries that simplify the problem.

**Derivatives of conditional marginals** In many probabilistic models, the marginal inference is a #P-hard problem (Roth, 1996), and this is not the case for the $k$-subset distribution. Theorem 1 in Ahmed et al. (2023) shows that the conditional marginals correspond to the partial derivatives of the log probability of the $k$-subset constraint. To see this, note that the derivative of a multi-linear function regarding a single variable retains all the terms referencing that variable and drops all other terms; this corresponds exactly to the unnormalized conditional marginals. By taking the derivative of the log probability, this introduces the $k$-subset probability in the denominator, leading to *conditional* marginals. Intuitively, the rate of change of the $k$-subset probability w.r.t. a variable only depends on that variable through its length-$k$ subsets. They further show in Proposition 1 in Ahmed et al. (2023) that the log probability of the exactly-$k$ constraint $p_{\boldsymbol{\theta}}(\sum_j z_j = k)$ is tractable is tractable as well as amenable to auto-differentiation, solving problem **(P1)** exactly and efficiently.

**Gradients of loss w.r.t. samples** What remains is estimating the loss value, requiring faithful sampling from the $k$-subset distribution. To perform exact sampling from the $k$-subset distribution, SIMPLE starts by sampling the variables in reverse order, that is, it samples $z_n$ through $z_1$. The intuition is that, having sampled $(z_n, z_{n-1}, \cdots, z_{i+1})$ with a Hamming weight of $k - j$, it samples $Z_i$ with a probability of choosing $k - j$ of $n - 1$ variables *and* the $n$th variable *given that* we choose $k - j + 1$ of $n$ variables, providing an exact and efficient solution to problem **(P2)**.

By combining the use of conditional marginal derivatives in the backward pass and the exact sampling in the forward pass, SIMPLE can achieve both low bias and low variance in its gradient estimation. We refer the readers to SIMPLE (Ahmed et al., 2023) for the proofs and full details of the approach.

## E DATASETS

Here, we give additional information regarding the datasets. The statistics of the datasets in our paper can be found in 5. Among them, ZINC, ALCHEMY, MUTAG, PTC_MR, NCI1, NCI109, PROTEINS, IMDB-B, and IMDB-M are from TUDatasets (Morris et al., 2020). Whereas PEPTIDES-FUNC and PEPTIDES-STRUCT are featured in Dwivedi et al. (2022b). Besides, CORNELL, TEXAS and WISCONSIN are WebKB datasets (Craven et al., 1998) also used in Pei et al. (2020). The OGB datasets are credited to Hu et al. (2020a). Moreover, we also incorporate synthetic datasets from the literature. EXP dataset consists of partially isomorphic graphs as described in Abboud et al. (2020), while the graphs in the CSL dataset are synthetic regular graphs proposed in Murphy et al. (2019). The construction of TREES-NEIGHBORSMATCH dataset is introduced in Alon & Yahav (2021).

Similar to the TREES-NEIGHBORSMATCH dataset, we propose our own TREES-LEAFCOUNT dataset. We fix a problem radius $R > 0$ and retrieve the binary representation of all numbers fitting into $2^R$ bits. This construction allows us to create $2^R$ unique binary trees by labeling the leaves with "0" and "1" corresponding to the binary equivalents of the numbers. A label is then assigned to the root node, reflecting the count of leaves tagged with "1". From the resulting graphs, we sample to ensure an equal

Table 5: Dataset statistics and properties for graph-level prediction tasks, [†]—Continuous vertex labels following Gilmer et al. (2017), the last three components encode 3D coordinates.

| DATASET | PROPERTIES | | | | | | |
|---|---|---|---|---|---|---|---|
| | NUMBER OF GRAPHS | NUMBER OF TARGETS | LOSS | ∅ NUMBER OF VERTICES | ∅ NUMBER OF EDGES | VERTEX LABELS | EDGE LABELS |
| ALCHEMY | 202 579 | 12 | MAE | 10.1 | 10.4 | ✓ | ✓ |
| QM9 | 129 433 | 13 | MAE | 18.0 | 18.6 | ✓(13+3D)[†] | ✓(4) |
| ZINC | 249 456 | 1 | MAE | 23.1 | 24.9 | ✓ | ✓ |
| EXP | 1 200 | 2 | ACC | 44.5 | 55.2 | ✓ | ✗ |
| CSL | 150 | 10 | ACC | 41.0 | 82.0 | ✗ | ✗ |
| OGBG-MOLHIV | 41 127 | 2 | ROCAUC | 25.5 | 27.5 | ✓ | ✓ |
| CORNELL | 1 | 5 | ACC | 183.0 | 298.0 | ✓ | ✗ |
| TEXAS | 1 | 5 | ACC | 183.0 | 325.0 | ✓ | ✗ |
| WISCONSIN | 1 | 5 | ACC | 251.0 | 515.0 | ✓ | ✗ |
| TREES-LEAFCOUNT($R = 4$) | 16 000 | 16 | ACC | 31 | 61 | ✓ | ✗ |
| TREES-NEIGHBORSMATCH($R = 4$) | 14 000 | 7 | ACC | 31 | 61 | ✓ | ✗ |
| PEPTIDES-FUNC | 15 535 | 10 | AP | 150.9 | 153.7 | ✓ | ✓ |
| PEPTIDES-STRUCT | 15 535 | 11 | MAE | 150.9 | 153.7 | ✓ | ✓ |
| MUTAG | 188 | 2 | ACC | 17.9 | 19.8 | ✓ | ✓ |
| PTC_MR | 344 | 2 | ACC | 14.3 | 14.7 | ✓ | ✓ |
| NCI1 | 4 110 | 2 | ACC | 29.9 | 32.3 | ✓ | ✗ |
| NCI109 | 4 127 | 2 | ACC | 29.7 | 32.1 | ✓ | ✗ |
| PROTEINS | 1 113 | 2 | ACC | 39.1 | 72.8 | ✓ | ✗ |
| IMDB-M | 1 500 | 3 | ACC | 13.0 | 65.9 | ✗ | ✗ |
| IMDB-B | 1 000 | 2 | ACC | 19.7 | 96.5 | ✗ | ✗ |

class distribution. The task requires a model to predict the root label, thereby requiring a strategy capable of conveying information from the leaves to the root.

We aim to have a controlled environment to observe if our upstream model $h_{\mathbf{u}}$ can sample meaningful edges for the new graph configuration. Conventionally, a minimum of $R$ message-passing layers is required to accomplish both tasks (Barceló et al., 2020; Alon & Yahav, 2021). However, a single-layer upstream MPNN could trivially resolve both datasets, provided the rewired graphs embed direct pathways from the root node to the leaf nodes containing the label information. To circumvent any potential bias within the sampling procedure, we utilize the self-attention mechanism described in Section 3 as our upstream model $h_{\mathbf{u}}$, along with a single-layer GIN architecture serving as the downstream model $f_{\mathbf{d}}$. For each problem radius, we sample exactly $k = 2^D$ edges. Indeed, our method consistently succeeded in correctly rewiring the graphs in all tested scenarios, extending up to a problem radius of $R = 6$, and achieved perfect test accuracy on both datasets. Figure 3 presents a qualitative result from the TREES-LEAFCOUNT dataset, further illustrating the capabilities of our approach.

## F   HYPERPARAMETER AND TRAINING DETAILS

**Experimental Protocol** Table 8 lists our hyperparameters choices. For all our experiments, we use early stopping with an initial learning rate of 0.001 that we decay by half on a plateau.

We compute each experiment's mean and standard deviation with different random seeds over a minimum of three runs. We take the best results from the literature for the other models, except for SAT on the OGBG-MOLHIV, where we use the same hyperparameters as the authors use on ZINC. We evaluate test predictive performance based on validation performance. In the case of the WEBKB datasets, we employ a 10-fold cross-validation with the provided data splits. For PEPTIDES, OGBG-MOLHIV, ALCHEMY, and ZINC, our models use positional and structural embeddings concatenated to the initial node features. Specifically, we add both RWSE and LAPPE (Dwivedi et al., 2022a). We use the same downstream model as the base model for the rewiring models.

Our code can be accessed at `https://github.com/chendiqian/PR-MPNN/`.

## G   ADDITIONAL EXPERIMENTAL RESULTS

Here, we report on the computation times of different variants of our probabilistic graph rewiring schemes and results on synthetic datasets.

**Training times** We report the average training time per epoch in Table 9. The RANDOM entry refers to using random adjacency matrices as rewired graphs.

**Extended TUDatasets** In addition to Table 4 in the main paper, we report the results of IMDB-B and IMDB-M datasets in Table 7. We also propose a proper train/validation/test splitting and show the results in Table 6.

Table 6: Extended results between our probabilistic rewiring method and the other approaches reported in Giusti et al. (2023b). Besides the 10-fold cross-validation, as in Giusti et al. (2023b); Xu et al. (2019), we provide a train/validation/test split. In addition, we also provide results on the IMDB-B and IMDB-M datasets. We use **green** for the best model, **blue** for the second-best, and **red** for third.

| MODEL | MUTAG | PTC_MR | PROTEINS | NCI1 | NCI109 | IMDB-B | IMDB-M |
|-------|-------|--------|----------|------|--------|--------|--------|
| PR-MPNN (10-FOLD CV) | **98.4**$_{\pm2.4}$ | **74.3**$_{\pm3.9}$ | **80.7**$_{\pm3.9}$ | **85.6**$_{\pm0.8}$ | **84.6**$_{\pm1.2}$ | **75.2**$_{\pm3.2}$ | **52.9**$_{\pm3.2}$ |
| PR-MPNN (TRAIN/VAL/TEST) | 91.0$_{\pm3.7}$ | 58.9$_{\pm5.0}$ | **79.1**$_{\pm2.8}$ | 81.5$_{\pm1.6}$ | 81.8$_{\pm1.5}$ | 71.6$_{\pm1.2}$ | 45.8$_{\pm0.8}$ |

Table 7: Extended comparison on the IMDB-B and IMDB-M datasets from the TUDATASET collection. We use **green** for the best model, **blue** for the second-best, and **red** for third.

| MODEL | IMDB-B | IMDB-M |
|-------|--------|--------|
| DGCNN (ZHANG ET AL., 2018) | 70.0$_{\pm0.9}$ | 47.8$_{\pm0.9}$ |
| IGN (MARON ET AL., 2019B) | 71.3$_{\pm4.5}$ | 48.6$_{\pm3.9}$ |
| GIN (XU ET AL., 2019) | 75.1$_{\pm5.1}$ | 52.3$_{\pm2.8}$ |
| PPGNs (MARON ET AL., 2019A) | 73.0$_{\pm5.7}$ | 50.4$_{\pm3.6}$ |
| NATURAL GN (DE HAAN ET AL., 2020) | 74.8$_{\pm2.0}$ | 51.2$_{\pm1.5}$ |
| GSN (BOURITSAS ET AL., 2022) | **77.8**$_{\pm3.3}$ | **54.3**$_{\pm3.3}$ |
| CIN (BODNAR ET AL., 2021) | **75.6**$_{\pm3.2}$ | **52.5**$_{\pm3.0}$ |
| PR-MPNN (10-FOLD CV) | **75.2**$_{\pm3.2}$ | **52.9**$_{\pm3.2}$ |
| PR-MPNN (TRAIN/VAL/TEST) | 71.6$_{\pm1.2}$ | 45.8$_{\pm0.8}$ |

**QM9** We compare our PR-MPNN with multiple current methods on QM9 dataset, see Table 10. The baselines are R-GNN in Alon & Yahav (2021), GNN-FiLM (Brockschmidt, 2020), SPN (Abboud et al., 2022) and the recent DRew paper (Gutteridge et al., 2023). Following the settings of Abboud et al. (2022) and Gutteridge et al. (2023), we train the network on each task separately. We use the normalized regression labels for training and report the de-normalized numbers. Similar to Abboud et al. (2022) and Gutteridge et al. (2023), we also exclude the 3D coordinates of the datasets. It is worth noting that our PR-MPNN reaches the overall lowest mean absolute error on HOMO, LUMO, gap, and Omega tasks while gaining at most $14.13\times$ better performance compared with a base GIN model.

**WebKB** To show PR-MPNNs's capability on heterophilic graphs, we carry out experiments on the three WebKB datasets, namely CORNELL, TEXAS, and WISCONSIN, Table 11. We compare with diffusion-based GNN (Gasteiger et al., 2019), Geom-GCN (Pei et al., 2020), and the recent graph rewiring work SDRF (Topping et al., 2021). Besides the MPNN baselines above, we also compare them against graph transformers. PR-MPNNs consistently outperform the other MPNN methods and are even better than graph transformers on the TEXAS dataset.

**LRGB** We apply PR-MPNNs on the two Long Range Graph Benchmark tasks (Dwivedi et al., 2022b), PEPTIDES-FUNC and PEPTIDES-STRUCT, which are graph classification and regression tasks, respectively. The baseline methods are also reported in Gutteridge et al. (2023). Notably, on PEPTIDES-STRUCT, PR-MPNNs reach the overall lowest mean absolute error.

## H ROBUSTNESS ANALYSIS

A beneficial side-effect of training PR-MPNNs is the enhanced robustness of the downstream model to graph structure perturbations. This is because PR-MPNNs generate multiple adjacency matrices for the same graph during training, akin to augmenting training data by randomly dropping or adding edges, but with a parametrized "drop/add" distribution rather than a uniform one.

To observe the performance degradation when testing on noisy data, we conduct an experiment on the PROTEINS dataset. After training our models on clean data, we compared the models' test accuracy on the clean and corrupted graphs. Corrupted graphs were generated by either deleting or adding a certain percentage of their edges. We report the change in the average test accuracy over 5 runs, comparing the base model with variants of PR-MPNN in Table 13.

Table 8: Overview of used hyperparameters.

| DATASET | HIDDEN$_{\text{UPSTREAM}}$ | HIDDEN$_{\text{DOWNSTREAM}}$ | LAYERS$_{\text{UPSTREAM}}$ | LAYERS$_{\text{DOWNSTREAM}}$ | K$_{\text{ADD}}$ | K$_{\text{BM}}$ | L$_{\text{ADD}}$ | HEUR | DROPOUT | N$_{\text{PRIORS}}$ | SAMPLES$_{\text{TRAIN/TEST}}$ |
|---|---|---|---|---|---|---|---|---|---|---|---|
| ZINC | {32, 64, 96} | 256 | {2, 4, 8} | 4 | [1, 128] | [1, 128] | 256 | DISTANCE | .0 | 1 | 5 |
| QM9 | 64 | {196, 256} | 8 | 4 | {5, 20, 35, 50, 80} | 5 | {100, 350} | DISTANCE | {.1, .5} | {1, 2} | {2, 3, 5} |
| PEPTIDES-FUNC | {128, 256} | {128, 256} | {4, 8} | {4, 8} | {16, 64, 256, 512} | {16, 64, 256, 512} | {128, 256, 512, 2048} | DISTANCE | {.0, .1} | {1, 2, 5} | {1, 2, 5} |
| PEPTIDES-STRUCT | {128, 256} | {128, 256} | {4, 8} | {4, 8} | {16, 64, 256, 512} | {16, 64, 256, 512} | {128, 256, 512, 2048} | DISTANCE | {.0, .1} | {1, 2, 5} | {1, 2, 5} |
| MUTAG | {32, 64} | {32, 64, 96} | {4, 8, 16} | {4, 8} | {0, 5, 10, 25} | {0, 5, 10, 25} | 256 | DISTANCE | {.0, .1, .2} | {2, 3} | {2, 5} |
| PTC_MR | {32, 64} | {32, 64, 96} | {4, 8, 16} | {4, 8} | {0, 5, 10, 25} | {0, 5, 10, 25} | 256 | DISTANCE | {.0, .1, .2} | {2, 3} | {2, 5} |
| NCI1 | {32, 64} | {32, 64, 96} | {4, 8, 16} | {4, 8} | {0, 5, 10, 25} | {0, 5, 10, 25} | 256 | DISTANCE | {.0, .1, .2} | {2, 3} | {2, 5} |
| NCI109 | {32, 64} | {32, 64, 96} | {4, 8, 16} | {4, 8} | {0, 5, 10, 25} | {0, 5, 10, 25} | 256 | DISTANCE | {.0, .1, .2} | {2, 3} | {2, 5} |
| PROTEINS | {32, 64} | {32, 64, 96} | {4, 8, 16} | {4, 8} | {0, 5, 10, 25} | {0, 5, 10, 25} | 256 | DISTANCE | {.0, .1, .2} | {2, 3} | {2, 5} |
| IMDB-M | {32, 64} | {32, 64, 96} | {4, 8, 16} | {4, 8} | {0, 5, 10, 25} | {0, 5, 10, 25} | 256 | DISTANCE | {.0, .1, .2} | {2, 3} | {2, 5} |
| IMDB-B | {32, 64} | {32, 64, 96} | {4, 8, 16} | {4, 8} | {0, 5, 10, 25} | {0, 5, 10, 25} | 256 | DISTANCE | {.0, .1, .2} | {2, 3} | {2, 5} |
| CORNELL | {64, 256} | 192 | {2, 3, 4} | 3 | [1024, 2048] | [256, 512] | 4096 | SIMILARITY | .0 | [1, 5] | [1, 5] |
| WISCONSIN | {64, 256} | 192 | {2, 3, 4} | 3 | [1024, 2048] | [256, 512] | 4096 | SIMILARITY | .0 | [1, 5] | [1, 5] |
| TEXAS | {64, 256} | 192 | {2, 3, 4} | 1 | [1024, 2048] | [256, 512] | 4096 | SIMILARITY | .0 | [1, 5] | [1, 5] |
| TREES-LEAFCOUNT | 32 | 32 | 1 | 1 | N$_{\text{LEAFS}}$ | ALL | - | ALL | .0 | 1 | 1 |
| TREES-NEIGHBORSMATCH | 32 | 32 | 2 | DEPTH+1 | {20, 32, 64, 128, 256} | 0 | - | ALL | .0 | 1 | 2 |
| CSL | 32 | 128 | {4, 8, 12} | 2 | 2 | 0 | 1 | DISTANCE | .0 | {1, 3, 5} | {1, 10} |
| 4-CYCLES | 16 | 16 | 4 | 4 | 4 | 2 | - | ALL | .0 | {1, 5, 10} | {5, 10, 20, 30, 40, 50} |
| EXP | 64 | 32 | 8 | 6 | {1, 5, 10, 15, 20, 25, 50, 100} | {1, 5, 10, 15, 20, 25, 50, 100} | 350 | DISTANCE | .0 | {1, 5, 10, 25} | {1, 5, 10} |

Table 9: Train and validation time per epoch in seconds for a GINE model, the SAT Graph Transformer, and PR-MPNN using different gradient estimators on OGBG-MOLHIV. The time is averaged over five epochs. PR-MPNN is approximately 5 times slower than a GINE model with a similar parameter count, while SAT is approximately 30 times slower than the GINE, and 6 times slower than the PR-MPNN models. Experiments performed on a machine with a single Nvidia RTX A5000 GPU and a Intel i9-11900K CPU.

| MODEL | #PARAMS | TOTAL SAMPLED EDGES | TRAIN TIME/EP (S) | VAL TIME/EP (S) |
|---|---|---|---|---|
| GINE | $502k$ | - | $3.19 \pm 0.03$ | $0.20 \pm 0.01$ |
| K-ST SAT$_{GINE}$ | $506k$ | - | $86.54 \pm 0.13$ | $4.78 \pm 0.01$ |
| K-SG SAT$_{GINE}$ | $481k$ | - | $97.94 \pm 0.31$ | $5.57 \pm 0.01$ |
| K-ST SAT$_{PNA}$ | $534k$ | - | $90.34 \pm 0.29$ | $4.85 \pm 0.01$ |
| K-SG SAT$_{PNA}$ | $509k$ | - | $118.75 \pm 0.50$ | $5.84 \pm 0.04$ |
| PR-MPNN$_{Gmb}$ | $582k$ | 20 | $15.20 \pm 0.08$ | $1.01 \pm 0.01$ |
| PR-MPNN$_{Gmb}$ | $582k$ | 100 | $18.18 \pm 0.08$ | $1.08 \pm 0.01$ |
| PR-MPNN$_{Imle}$ | $582k$ | 20 | $15.01 \pm 0.22$ | $1.08 \pm 0.06$ |
| PR-MPNN$_{Imle}$ | $582k$ | 100 | $15.13 \pm 0.17$ | $1.13 \pm 0.06$ |
| PR-MPNN$_{Sim}$ | $582k$ | 20 | $15.98 \pm 0.13$ | $1.07 \pm 0.01$ |
| PR-MPNN$_{Sim}$ | $582k$ | 100 | $17.23 \pm 0.15$ | $1.15 \pm 0.01$ |

Table 10: Performance of PR-MPNN on QM9, in comparison with the base downstrem model (Base-GIN) and other competing methods. The relative improvement of PR-MPNN over the base downstream model is reported in the paranthesis. The metric used is MAE, lower scores are better. We note the best performing method with **green**, the second-best with **blue**, and third with **orange**.

| PROPERTY | R-GIN+FA | GNN-FILM | SPN | DRew-GIN | BASE-GIN | PR-MPNN |
|---|---|---|---|---|---|---|
| MU | $2.54 \pm 0.09$ | $2.38 \pm 0.13$ | $2.32 \pm 0.28$ | $1.93 \pm 0.06$ | $2.64 \pm 0.01$ | $1.99 \pm 0.02$ (1.33×) |
| ALPHA | $2.28 \pm 0.04$ | $3.75 \pm 0.11$ | $1.77 \pm 0.09$ | $1.63 \pm 0.03$ | $7.67 \pm 0.16$ | $2.28 \pm 0.06$ (3.36×) |
| HOMO | $1.26 \pm 0.02$ | $1.22 \pm 0.07$ | $1.26 \pm 0.09$ | $1.16 \pm 0.01$ | $1.70 \pm 0.02$ | $1.14 \pm 0.01$ (1.49×) |
| LUMO | $1.34 \pm 0.04$ | $1.30 \pm 0.05$ | $1.19 \pm 0.05$ | $1.13 \pm 0.02$ | $3.05 \pm 0.01$ | $1.12 \pm 0.01$ (2.72×) |
| GAP | $1.96 \pm 0.04$ | $1.96 \pm 0.06$ | $1.89 \pm 0.11$ | $1.74 \pm 0.02$ | $3.37 \pm 0.03$ | $1.70 \pm 0.01$ (1.98×) |
| R2 | $12.61 \pm 0.37$ | $15.59 \pm 1.38$ | $10.66 \pm 0.40$ | $9.39 \pm 0.13$ | $23.35 \pm 1.08$ | $10.41 \pm 0.35$ (2.24×) |
| ZPVE | $5.03 \pm 0.36$ | $11.00 \pm 0.74$ | $2.77 \pm 0.17$ | $2.73 \pm 0.19$ | $66.87 \pm 1.45$ | $4.73 \pm 0.08$ (14.13×) |
| U0 | $2.21 \pm 0.12$ | $5.43 \pm 0.96$ | $1.12 \pm 0.13$ | $1.01 \pm 0.09$ | $21.48 \pm 0.17$ | $2.23 \pm 0.13$ (9.38×) |
| U | $2.32 \pm 0.18$ | $5.95 \pm 0.46$ | $1.03 \pm 0.09$ | $0.99 \pm 0.08$ | $21.59 \pm 0.30$ | $2.31 \pm 0.06$ (9.35×) |
| H | $2.26 \pm 0.19$ | $5.59 \pm 0.57$ | $1.05 \pm 0.04$ | $1.06 \pm 0.09$ | $21.96 \pm 1.24$ | $2.66 \pm 0.01$ (8.26×) |
| G | $2.04 \pm 0.24$ | $5.17 \pm 1.13$ | $0.97 \pm 0.06$ | $1.06 \pm 0.14$ | $19.53 \pm 0.47$ | $2.24 \pm 0.01$ (8.24×) |
| Cv | $1.86 \pm 0.03$ | $3.46 \pm 0.21$ | $1.36 \pm 0.06$ | $1.24 \pm 0.02$ | $7.34 \pm 0.06$ | $1.44 \pm 0.01$ (5.10×) |
| OMEGA | $0.80 \pm 0.04$ | $0.98 \pm 0.06$ | $0.57 \pm 0.04$ | $0.55 \pm 0.01$ | $0.60 \pm 0.03$ | $0.48 \pm 0.00$ (1.25×) |

Table 11: Quantitative results on the heterophilic and transductive WEBKB datasets. **Best overall**; **Second best**; **Third best**. Rewiring outperforms the base models on all of the datasets. Graph transformers have an advantage over both the base models and the ones employing rewiring.

| | | HETEROPHILIC & TRANSDUCTIVE | | |
| | | CORNELL ↑ | TEXAS ↑ | WISCONSIN ↑ |
|---|---|---|---|---|
| MPNNs | BASE | 0.574±0.006 | 0.674±0.010 | 0.697±0.013 |
| | BASE w. PE | 0.540±0.043 | 0.654±0.010 | 0.649±0.018 |
| | DIGL (GASTEIGER ET AL., 2019) | 0.582±0.005 | 0.620±0.003 | 0.495±0.003 |
| | DIGL + UNDIRECTED (GASTEIGER ET AL., 2019) | 0.595±0.006 | 0.635±0.004 | 0.522±0.005 |
| | GEOM-GCN (PEI ET AL., 2020) | 0.608± N/A | 0.676± N/A | 0.641± N/A |
| | SDRF (TOPPING ET AL., 2021) | 0.546±0.004 | 0.644±0.004 | 0.555±0.003 |
| | SDRF + UNDIRECTED (TOPPING ET AL., 2021) | 0.575±0.003 | 0.703±0.006 | 0.615±0.008 |
| | PR-MPNN (OURS) | 0.659±0.040 | **0.827±0.032** | 0.750±0.015 |
| GTs | GPS (LAPPE) | 0.662±0.038 | **0.778±0.010** | 0.747±0.029 |
| | GPS (RWSE) | **0.708±0.020** | **0.775±0.012** | **0.802±0.022** |
| | GPS (DEG) | **0.718±0.024** | 0.773±0.013 | **0.798±0.090** |
| | GRAPHORMER (DEG) | **0.683±0.017** | 0.767±0.017 | **0.770±0.019** |
| | GRAPHORMER (DEG + ATTN BIAS) | **0.683±0.017** | 0.767±0.017 | **0.770±0.019** |

Table 12: Comparison between PR-MPNN and other methods as reported in Gutteridge et al. (2023). **Best overall**; **Second best**; **Third best**. PR-MPNN obtains the best score on the PEPTIDES-STRUCT dataset from the LRGB collection, but ranks below Drew on the PEPTIDES-FUNC dataset.

| MODEL | PEPTIDES-FUNC AP ↑ | PEPTIDES-STRUCT MAE ↓ |
|---|---|---|
| GCN | 0.5930±0.0023 | 0.3496±0.0013 |
| GINE | 0.5498±0.0079 | 0.3547±0.0045 |
| GATEDGCN | 0.5864±0.0077 | 0.3420±0.0013 |
| GATEDGCN+PE | 0.6069±0.0035 | 0.3357±0.0006 |
| DIGL+MPNN | 0.6469±0.0019 | 0.3173±0.0007 |
| DIGL+MPNN+LAPPE | 0.6830±0.0026 | 0.2616±0.0018 |
| MIXHOP-GCN | 0.6592±0.0036 | 0.2921±0.0023 |
| MIXHOP-GCN+LAPPE | 0.6843±0.0049 | 0.2614±0.0023 |
| TRANSFORMER+LAPPE | 0.6326±0.0126 | **0.2529±0.0016** |
| SAN+LAPPE | 0.6384±0.0121 | 0.2683±0.0043 |
| GRAPHGPS+LAPPE | 0.6535±0.0041 | **0.2500±0.0005** |
| DREW-GCN | 0.6996±0.0076 | 0.2781±0.0028 |
| DREW-GCN+LAPPE | **0.7150±0.0044** | 0.2536±0.0015 |
| DREW-GIN | 0.6940±0.0074 | 0.2799±0.0016 |
| DREW-GIN+LAPPE | **0.7126±0.0045** | 0.2606±0.0014 |
| DREW-GATEDGCN | 0.6733±0.0094 | 0.2699±0.0018 |
| DREW-GATEDGCN+LAPPE | **0.6977±0.0026** | 0.2539±0.0007 |
| PR-MPNN | 0.6825±0.0086 | **0.2477±0.0005** |

Table 13: Robustness results on the PROTEINS dataset when testing on various levels of noise, obtained by removing or adding random edges. The percentages indicate the change in the average test accuracy over 5 runs. PR-MPNNs consistently obtain the best results for both removing $k = 25$ and $k = 50$ edges.

| MODEL NOISE | GINE | PR-MPNN$_{k=25}$ | PR-MPNN$_{k=50}$ |
|---|---|---|---|
| RM 10% | -7.68% | +0.04% | +0.34% |
| RM 30% | -11.87% | -2.56% | -0.56% |
| RM 50% | -9.18% | -3.50% | -0.21% |
| ADD 10% | -5.93% | +0.28% | +0.80% |
| ADD 30% | -11.14% | -1.27% | +0.28% |
| ADD 50% | -21.75% | -1.99% | -0.43% |

