# OpenReview forum: "Probabilistically Rewired Message-Passing Neural Networks"
_ICLR.cc/2024/Conference — ICLR 2024 poster_

### Official Review · Reviewer_j89x · 2023-10-29

**Soundness:** 3 good
**Presentation:** 3 good
**Contribution:** 3 good
**Rating:** 8
**Confidence:** 4

**Summary:**

The work proposes a probabilistic rewiring technique that relies on differentiable k-subset sampling. The main motivation is that while current rewiring techniques rely on "arbitrary" heuristics (improving spectral gap, connecting nodes in the 2-hop), it would be ideal to *learn* such a rewiring. Further, the work proposes a way to study the expressive power -- in terms of WL -- for such a process. The work is evaluated on synthetic and real-world benchmarks.

**Strengths:**

The work is well-written and overall I found it interesting. I agree that learning a rewiring is intuitively something that should be preferred over arbitrary heuristics. I also appreciated that the work discusses techniques such as the Differentiable Graph Module, which in principle may seem similar.

**Weaknesses:**

(W1) I believe experimentally there could be a larger breadth of rewiring benchmarks, for instance comparing against "deterministic" rewiring techniques such as FOSR [1] and SDRF [2] would be valuable. Furthermore, adding a "random rewiring" GNN, i.e. DropGNN, would also be useful for the real-world tasks.

(W2) The claim that the rewiring technique reduces over-squashing could be strengthened. At the moment this seems to be solely motivated by the empirical results in Figure 2 and Figure 4.

[1] FoSR: First-order spectral rewiring for addressing oversquashing in GNNs. Kedar Karhadkar, Pradeep Kr. Banerjee, Guido Montúfar. ICLR 2023

[2] Jake Topping, Francesco Di Giovanni, Benjamin Paul Chamberlain, Xiaowen Dong, and Michael M Bronstein. Understanding over-squashing and bottlenecks on graphs via curvature. ICLR 2022

**Questions:**

(Q1) From Figure 1 and the paper overall, it seems like the rewiring technique is only removing edges and not adding any new edges not present in the original graph. Is this the case?

(Q2) Regarding (W1), would it be possible to show results for existing deterministic rewiring techniques?

(Q3) Regarding (W2) and especially (Q1), it is not clear to me how the technique can reduce over-squashing if it is only able to remove edges. It would be important to clarify (Q1), and provide some theoretical evidence that it is indeed able to alleviate over-squashing. In general by removing edges, one is reducing the total effective resistance over the graph.

---

> ### Author Response · Authors · 2023-11-16
>
> We thank the reviewer for their helpful comments and observations. We will respond to their raised issues in the following:
>
> __W1__: More comparisons with deterministic rewiring methods.
> __RW1__: Thank you for your suggestion. In our original manuscript, comparisons with deterministic rewiring methods such as DIGL and SDRF on heterophilic datasets were presented in Supplementary Table 11, with Drew on the QM9 dataset in Supplementary Table 10, and with DIGL and Drew on the Peptides datasets in Supplementary Table 12. Following the reviewer's recommendation, we have now included additional comparisons with FoSR and DropGNN on the TUDataset datasets in Table 4 and have expanded our analysis to cover all variants of SDRF and DIGL on the Heterophilic datasets in Supplementary Table 12.
>
> __W2__: Over-squashing claim could be strengthened.
> __RW2__: We thank the reviewer for their suggestion.  Our empirical analysis of the widely-used Trees-NeighborsMatch dataset [1,2,3,4] demonstrates how PR-MPNNs alleviate over-squashing. As illustrated in Figure 4, PR-MPNNs outperform standard MPNNs and match Graph Transformers' performance, which is believed to not suffer from over-squashing issues [2].
>
> We appreciate the suggestion and agree that further exploring the over-squashing phenomena with PR-MPNNs could be insightful. However, our experience indicates that traditional analyses using topological properties, as in references [3, 5], are not entirely adequate for black-box rewiring methods like ours. Despite logging various topological features such as edge curvatures and spectral gaps, no significant patterns emerged in the rewired graphs.
>
> To illustrate why we believe that topological and spectral properties may be limited when employing learnable rewiring methods, we give an intuition via the following example:
>
> [image1.png](https://postimg.cc/fkBDgqgX)
>
> Suppose that, to solve some task, nodes A and B from the initial graph a) need to pass messages, but there’s an information bottleneck between C and D. Increasing curvature in the rewiring process (as in example b) might improve information flow. However, a trivial solution, that could be obtained with PR-MPNNs, is to connect nodes A and B, bypassing the bottleneck directly. Moreover, if we delete the edge connecting nodes C and D, we obtain a graph that has the same curvature as the initial graph, as exemplified in example c).
>
> This simplified example may not fully capture the complexity encountered in real-world scenarios. However, we want to point out that PR-MPNNs are indeed able to find perfect solutions in this kind of toy scenarios, as illustrated in Figure 3, where the PR-MPNN is recovering the trivial solution to the task for the Trees-LeafCount synthetic dataset.
>
> [1]: Alon, Uri, and Eran Yahav. "On the bottleneck of graph neural networks and its practical implications." arXiv preprint arXiv:2006.05205 (2020).
> [2]: Müller, Luis, et al. "Attending to graph transformers." arXiv preprint arXiv:2302.04181 (2023).
> [3]: Karhadkar, Kedar, Pradeep Kr Banerjee, and Guido Montúfar. "FoSR: First-order spectral rewiring for addressing oversquashing in GNNs." arXiv preprint arXiv:2210.11790 (2022).
> [4]: Banerjee, Pradeep Kr, et al. "Oversquashing in GNNs through the lens of information contraction and graph expansion." 2022 58th Annual Allerton Conference on Communication, Control, and Computing (Allerton). IEEE, 2022.
> [5]: Topping, Jake, et al. "Understanding over-squashing and bottlenecks on graphs via curvature." arXiv preprint arXiv:2111.14522 (2021).
>
> __Q1__: Are new edges being added when rewiring?
> __RQ1__: Thank you for highlighting this aspect. In our paper, while we focus on edge removal in Figure 5 concerning theoretical discussion in relation to randomized methods like DropEdge and DropGNN, in practical applications, we perform both edge addition and deletion. Detailed statistics on the number of edges added and removed are provided in Supplementary Table 8. To enhance clarity, we have revised our manuscript to explicitly state that we are both adding and removing edges.
>
> __Q2__: More comparisons with deterministic rewiring methods.
> __RQ2__: Please see our response to Weakness 1.
>
> __Q3__: Evidence that we reduce over-squashing.
> __RQ3__: As noted in our response to Question 1, our method involves both edge addition and deletion. For a detailed discussion on why spectral and topological analysis may not adequately address over-squashing in the context of learnable rewiring, please refer to our response to Weakness 2. We posit this is also the case for methods that treat graphs as fully-connected, such as Graph Transformers.

---

> ### Comment · Reviewer_j89x · 2023-11-17
>
> I would like to thank the authors for the reply. I think this is a nice piece of work and recommend acceptance to the conference.

---

> ### Author Response · Authors · 2023-11-17
>
> We appreciate your kind reply. We would kindly ask you to adjust your score if we have addressed the concerns. Please also let us know what we can further improve.

---

> > ### Comment · Reviewer_j89x · 2023-11-20
> >
> > After further careful thought, I agree that the authors have been thorough in the rebuttal and more generally in their work. I have increased the score accordingly.

---

### Official Review · Reviewer_8vRc · 2023-11-01

**Soundness:** 3 good
**Presentation:** 2 fair
**Contribution:** 3 good
**Rating:** 6
**Confidence:** 4

**Summary:**

The paper proposes a probabilistic rewired message-passing network (PR-MPNN) to address the under-reaching and over-squashing problems of existing graph neural network (GNN) models. Specifically, PR-MPNN first uses a GNN to learn the priors over edges, and then it samples multiple adjacency matrices from the edge prior distributions using SIMPLE, a gradient estimator for k-subset sampling. The sampled adjacency matrices are combined with the original adjacency matrix to obtain the rewired graph, which is used in the downstream tasks. The paper also provides theoretical analysis to identify conditions under which the proposed method can outperform randomized approaches.

**Strengths:**

1. The proposed PR-MPNN model is simple yet effective.
2. The paper provides theoretical analyses to prove that the proposed model is more effective in probabilistically separating graphs than randomized approaches such as dropping nodes and edges uniformly under certain conditions.
3. Experimental results on both node classification and graph classification tasks indicate that PR-MPNN can achieve better or competitive performance compared with baselines.

**Weaknesses:**

1. The motivation for using $k$-subset constraint when sampling the adjacency matrix is not very clear. What are the advantages of using such constraints?

2. The difference between the proposed PR-MPNN and previous works is not explicitly discussed in the related work section.

3. The theoretical results indicate that PR-MPNN  is more effective in probabilistically separating graphs than randomized approaches under certain conditions. But how does this help PR-MPNN address the over-squashing problem?

4. In the introduction section, the paper states that PR-MPNNs make MPNNs less vulnerable to potential noise and missing information. However, there are no empirical or theoretical results to validate such statements.

5. PR-MPNN use SIMPLE to sample adjacency matrices. It seems to be an important component of the proposed model and the paper should provide an introduction to the SIMPLE method to make the paper self-contained.

6. I also have some concerns regarding the experiments:

(1) In the “Baseline and model configurations” paragraph, the paper states that there are two ways to leverage the sampled adjacency matrices when using multiple priors. However, it is unclear which method is used in their experiments.

(2) When answering Q2, why compare PR-MPNN with different baselines on different datasets? For example, the paper compares PR-MPNN with OSAN, GIN+POSENC, DropGNN on EXP, CSL and 4-CYCLES datasets respectively. Also, the statement “Concerning Q2, on the 4-CYCLES dataset, our probabilistic rewiring method consistently outperforms DropGNN” is inaccurate since PR-MPNN only achieve comparable performance with DropGNN in some cases.

(3) The results in Table 1 are quite confusing. What evaluation metric is used in these results? Why is it that on some datasets, such as OGBG-MOLHIV, a higher metric indicates better performance, while on others, such as ZINC and ALCHEMY,  a lower value is preferable?

(4) There are no ablation studies to validate the effectiveness of each component of the proposed model.

7. In the conclusion section, the paper states that PR-MPNN “is competitive or superior to conventional  MPNN models and graph transformer architectures regarding predictive performance and computational efficiency”. However, there is no comparison between PR-MPNN and baselines regarding the computational efficiency in the main text.

**Questions:**

Please see the questions in the Weaknesses section.

---

> ### Author Response · Authors · 2023-11-16
>
> We thank the reviewer for their thorough feedback. In the following, we address the raised issues one by one.
>
> __W1__: The motivation for k-subset sampling is not clear.
> __RW1__: The choice of sampling $k$-subsets with $k>1$ is required for introducing complex dependencies between the edge random variables. On the other hand, even more complex constraints on these random variables would render the computation of marginals, as needed for the gradient estimation algorithm SIMPLE, intractable. Moreover, the $k$-subset sampling also enables fine-grained control of the sampled graphs’ sparsity. In the final paper, we will add a detailed discussion.
>
> __W2__: The difference with prior work is not discussed.
> __RW2__: We appreciate the reviewer's remark and have made improvements to the related work section, clarifying our method's difference from existing heuristic and probabilistic rewiring techniques. Our manuscript's "Related Work" section extensively contrasts PR-MPNNs with heuristic-based methods such as DIGL, SDRF [1], SP-MPNN [2], and Drew [3]. We emphasize the novelty of our learning-based rewiring approach and its robust theoretical link with WL expressivity. Additionally, and as noted by Reviewer j89x, we want to point to the extended section in the Appendix, which addressed the relation to Graph Structure Learning (GSL) methods, which bear conceptual similarities to our work but typically employ $k$-NN algorithms for sparsification or model edges with independent Bernoulli variables. Moreover, the GSL methods mainly focus on node classification tasks, while we’re more interested in graph-level data. Nevertheless, our theoretical PR-MPNN framework extends to various GSL methods, offering broader applicability.
>
> [1]: Topping, Jake, et al. "Understanding over-squashing and bottlenecks on graphs via curvature." arXiv preprint arXiv:2111.14522 (2021).
> [2]: Abboud, Ralph, Radoslav Dimitrov, and Ismail Ilkan Ceylan. "Shortest path networks for graph property prediction." Learning on Graphs Conference. PMLR, 2022.
> [3]: Gutteridge, Benjamin, et al. "Drew: Dynamically rewired message passing with delay." International Conference on Machine Learning. PMLR, 2023.
>
> __W3__: How does PR-MPNN help alleviate over-squashing?
> __RW3__: Our empirical analysis on the widely-used Trees-NeighborsMatch dataset [1,2,3,4] demonstrates how PR-MPNNs alleviate over-squashing. As illustrated in Figure 4, PR-MPNNs outperform standard MPNNs and match Graph Transformers' performance, which is believed to suffer less from over-squashing issues [2]. Using these benchmark datasets is the de facto community standard for assessing the over-squashing behavior of MPNNs.
>
> [1]: Alon, Uri, and Eran Yahav. "On the bottleneck of graph neural networks and its practical implications." arXiv preprint arXiv:2006.05205 (2020).
> [2]: Müller, Luis, et al. "Attending to graph transformers." arXiv preprint arXiv:2302.04181 (2023).
> [3]: Karhadkar, Kedar, Pradeep Kr Banerjee, and Guido Montúfar. "FoSR: First-order spectral rewiring for addressing oversquashing in GNNs." arXiv preprint arXiv:2210.11790 (2022).
> [4]: Banerjee, Pradeep Kr, et al. "Oversquashing in GNNs through the lens of information contraction and graph expansion." 2022 58th Annual Allerton Conference on Communication, Control, and Computing (Allerton). IEEE, 2022.

---

> ### Author Response · Authors · 2023-11-16
>
> __W4__: No experiments to validate claims about robustness.
> __RW4__: We thank the reviewer for this observation. A beneficial side-effect of training PR-MPNNs is the enhanced robustness of the downstream model to graph structure perturbations. This is because PR-MPNNs generate multiple adjacency matrices for the same graph during training, akin to augmenting training data by randomly dropping or adding edges, but with a parametrized "drop/add" distribution rather than a uniform one.
>
> Motivated by the reviewer’s feedback, we conducted a dedicated experiment on the PROTEINS dataset. After training our models on clean data, we compared the models’ test accuracy on the clean and corrupted graphs. Corrupted graphs were generated by either deleting or adding a certain percentage of their edges. We report the change in the average test accuracy over 5 runs, comparing the base model with variants of PR-MPNN in the following table:
>
> | Corruption\\Model | GINE (acc 77.9) | PR-MPNN-RM25 (acc 80.1) | PR-MPNN-RM50 (acc 78.8) |
> | ----------------- | --------------- | ----------------------- | ------------------------ |
> | RM 10%            | \-7.68%         | +0.04%                  | __+0.34%__                   |
> | RM 30%            | \-11.87%        | \-2.56%                 | __\-0.56%__                  |
> | RM 50%            | \-9.18%         | \-3.50%                 | __\-0.21%__               |
> | ADD 10%           | \-5.93%         | +0.28%                  | __+0.80%__                   |
> | ADD 30%           | \-11.14%        | \-1.27%                 | __+0.28%__                   |
> | ADD 50%           | \-21.75%        | \-1.99%                 | __\-0.43%__                  |
>
> PR-MPNNs consistently obtain the best results for both removing k=25 and k=50 edges. The three models achieve test accuracies on the unperturbed dataset as follows: GINE 77.9%, PR-MPNN-RM25 80.1%, and PR-MPNN-RM50 78.8%.
> We included these initial results in the revised manuscript; see Table 13 in the appendix.
>
> __W5__: SIMPLE does not have a detailed explanation.
> __RW5__: We thank the reviewer for their remark. We have added an explanation of SIMPLE to the appendix of the updated submission.
>
> __W6__: Concerns about experiments.
> __(1)__ Confusion about how we leverage the adjacency matrices.
> __R__: We thank the reviewer for the remark. To clarify, we always use a downstream ensemble to obtain the results when using multiple rewired adjacency matrices. We have updated our manuscript with a clarification.
>
> __(2)__ a) Different baselines for different datasets.
> __R__: We selected different baselines for each dataset based primarily on the availability of the authors’ code and published results. Our evaluation of the PR-MPNN method covers 20 datasets, detailed in Table 5 of the Appendix. Due to time and resource limitations, it wasn't feasible to comprehensively evaluate all baselines across these datasets. We relied on publicly available results for comparisons where the direct evaluation was impractical. However, we conducted additional experiments for some baselines on datasets that were not initially reported (e.g., SAT on OGBG-Molhiv, GIN, and GIN+PE in Tables 1, 3, 10, and 11). Furthermore, we have introduced two new baselines on the TUDataset and have run new experiments with FoSR (PTC_MR, NCI1, NCI109) and DropGNN (NCI1, NCI109).
> __(2)__ b) Comparable performance with DropGNN on 4-Cycles.
> __R__: We appreciate the reviewer pointing this out. We intended to compare PR-MPNNs with DropGNN on 4-Cycles to emphasize the significance of utilizing multiple priors in sampling. This key aspect is detailed in the caption of Figure 2, where we illustrate the impact of our sampling approach. We have updated the text in the revised manuscript to reflect this.
>
> __(3)__ Table 1 is confusing.
> __R__: We appreciate the reviewer's feedback on Table 1. The evaluation metric for the ZINC and Alchemy datasets is the Mean Absolute Error (MAE), with a lower score indicating better performance. For the OGBG-Molhiv dataset, the metric is ROCAUC, where a higher score denotes superior performance. In our initial submission, we used symbols $\uparrow$ and $\downarrow$ to signify this. In our revised version, we explicitly state the evaluation metric for each dataset in Supplementary Table 5 to improve clarity.
>
> __(4)__ There are no ablation studies.
> __R__: We have conducted ablation studies focusing on the performance impact of different gradient estimators, specifically Gumbel, I-MLE, and SIMPLE, as detailed in Table 1. Additionally, our comparison with DropGNN, presented in Figure 2, demonstrates the benefits of utilizing multiple priors and samples. Furthermore, we have revised the computation efficiency in Table 9 in the Appendix, where we compare the runtime of various PR-MPNN models when using different gradient estimators and the total number of edges sampled. These sections collectively serve as our approach's ablation analysis.

---

> ### Author Response · Authors · 2023-11-16
>
> __W7__: No computational efficiency in the main text.
> __RW7__: We compute the computational complexity of PR-MPNNs in Section 3, noting that we obtain a better worst-case complexity than full self-attention-based graph transformers in the downstream module during test time, i.e., $\mathcal{O}(n^2)$ for a graph transformer versus $\mathcal{O}(L)$ for PR-MPNNs, where $n$ is the number of nodes in the graph, and $L$ is the number of edges in the rewired graph. Moreover, we also indicate that we can further reduce the complexity during training by restricting the edge candidates set, therefore achieving a lower overall complexity than a graph transformer. In the following table, we compare our runtime with a simple GINE model and the SAT Graph Transformer. PR-MPNN is approximately 5 times slower than a GINE model with a similar parameter count, while SAT is approximately 30 times slower than the GINE, and 6 times slower than the PR-MPNN models.
>
> | Model           | #Params | Total sampled edges | Train time/ep (s) | Val time/ep (s) |
> |-----------------|---------|---------------------|-------------------|-----------------|
> | GINE            | 502k    | -                   | 3.19 ± 0.03       | 0.20 ± 0.01     |
> | K-ST SAT (GINE) | 506k    | -                   | 86.54 ± 0.13      | 4.78 ± 0.01     |
> | K-SG SAT (GINE) | 481k    | -                   | 97.94 ± 0.31      | 5.57 ± 0.01     |
> | K-ST SAT (PNA)  | 534k    | -                   | 90.34 ± 0.29      | 4.85 ± 0.01     |
> | K-SG SAT (PNA)  | 509k    | -                   | 118.75 ± 0.50     | 5.84 ± 0.04     |
> | PR-MPNN (Gmb)   | 582k    | 20                  | 15.20 ± 0.08      | 1.01 ± 0.01     |
> | PR-MPNN (Gmb)   | 582k    | 100                 | 18.18 ± 0.08      | 1.08 ± 0.01     |
> | PR-MPNN (IMLE)  | 582k    | 20                  | 15.01 ± 0.22      | 1.08 ± 0.06     |
> | PR-MPNN (IMLE)  | 582k    | 100                 | 15.13 ± 0.17      | 1.13 ± 0.06     |
> | PR-MPNN (Sim)   | 582k    | 20                  | 15.98 ± 0.13      | 1.07 ± 0.01     |
> | PR-MPNN (Sim)   | 582k    | 100                 | 17.23 ± 0.15      | 1.15 ± 0.01     |
>
> Furthermore, we have also revised Table 9 in the Appendix.

---

> > ### Comment · Reviewer_8vRc · 2023-11-23
> >
> > I would like to thank the authors for their responses. I think the responses have addressed most of my concerns. Therefore, I have raised the score.

---

### Official Review · Reviewer_MyQT · 2023-11-03

**Soundness:** 3 good
**Presentation:** 3 good
**Contribution:** 3 good
**Rating:** 6
**Confidence:** 2

**Summary:**

The paper has proposed a probabilistic rewiring methanism for GNN. Its improvement in expressive power has been verified with theory and experiments.

**Strengths:**

1. The method has been verified with both theory and experiments.

**Weaknesses:**

1. The computation overhead is also needed.
2. How does the expressiveness guarantee translate to practical performances?

**Questions:**

I wonder what is the performance gain over the overhead the method produces.

**Details Of Ethics Concerns:**

No concern

---

> ### Author Response · Authors · 2023-11-16
>
> We thank the reviewer for their helpful comments and the positive evaluation of our work.
>
> __W1__: The computation overhead is also needed.
> __RW1__: We compute the computational complexity of PR-MPNNs in Section 3, noting that we obtain a better worst-case complexity than full self-attention-based graph transformers in the downstream module during test time, i.e., $\mathcal{O}(n^2)$ for a graph transformer versus $\mathcal{O}(L)$ for PR-MPNNs, where $n$ is the number of nodes in the graph, and $L$ is the number of edges in the rewired graph. Moreover, we also indicate that we can further reduce the complexity during training by restricting the edge candidates set, therefore achieving a lower overall complexity than a graph transformer. In the following table, we compare our runtime with a simple GINE model and the SAT Graph Transformer. PR-MPNN is approximately 5 times slower than a GINE model with a similar parameter count, while SAT is approximately 30 times slower than the GINE, and 6 times slower than the PR-MPNN models.
>
> | Model           | #Params | Total sampled edges | Train time/ep (s) | Val time/ep (s) |
> |-----------------|---------|---------------------|-------------------|-----------------|
> | GINE            | 502k    | -                   | 3.19 ± 0.03       | 0.20 ± 0.01     |
> | K-ST SAT (GINE) | 506k    | -                   | 86.54 ± 0.13      | 4.78 ± 0.01     |
> | K-SG SAT (GINE) | 481k    | -                   | 97.94 ± 0.31      | 5.57 ± 0.01     |
> | K-ST SAT (PNA)  | 534k    | -                   | 90.34 ± 0.29      | 4.85 ± 0.01     |
> | K-SG SAT (PNA)  | 509k    | -                   | 118.75 ± 0.50     | 5.84 ± 0.04     |
> | PR-MPNN (Gmb)   | 582k    | 20                  | 15.20 ± 0.08      | 1.01 ± 0.01     |
> | PR-MPNN (Gmb)   | 582k    | 100                 | 18.18 ± 0.08      | 1.08 ± 0.01     |
> | PR-MPNN (IMLE)  | 582k    | 20                  | 15.01 ± 0.22      | 1.08 ± 0.06     |
> | PR-MPNN (IMLE)  | 582k    | 100                 | 15.13 ± 0.17      | 1.13 ± 0.06     |
> | PR-MPNN (Sim)   | 582k    | 20                  | 15.98 ± 0.13      | 1.07 ± 0.01     |
> | PR-MPNN (Sim)   | 582k    | 100                 | 17.23 ± 0.15      | 1.15 ± 0.01     |
>
> Furthermore, we have also revised Table 9 in the Appendix.
>
> __W2__: How does the expressiveness translate to practical performance?
> __RW2__: Our empirical results validate the expressivity claims using the EXP and CSL datasets. These datasets are designed so GNNs limited to 1-WL expressivity will not perform well. As detailed in Tables 2 and 3, our rewiring technique achieves perfect performance on EXP and nearly perfect on CSL, significantly outperforming the base GIN model's random performance, clearly indicating that our expressivity guarantees translate into practice.
>
> In addition, as shown in Table 1, our SIMPLE-based rewiring method consistently surpasses the base model regarding predictive performance on real-world datasets. Further, Table 10 highlights a substantial improvement, up to 14.13x, over the base GIN model on QM9, demonstrating significant real-world performance gains.

---

### Author Response · Authors · 2023-11-16
**Global response**

We thank all reviewer for their fair and constructive reviews. We thoroughly revised the paper, taking into account the reviewers’ suggestions. In the following, we summarize our changes and additions.

- We have improved the related work section, clarifying the difference between our method to existing heuristic and probabilistic rewiring techniques.
- We clarified that we can both learn to remove and add edges.
- In the experimental setup, we made clear that we always used downstream ensemble when sampling multiple adjacency matrix
- In the answers below, we added a more detailed discussion on how PR-MPNNs alleviate over-squashing.
- In the answers below, we added a more detailed discussion on the benefits of k-subset sampling; we will add this to the final paper.
- In Appendix D, we added an introduction to SIMPLE
- To compare to further baselines, we compared to the DropGNN model (Papp et al., 2021) and FOSR (Karhadkar et al., 20223); see Table 4. The results clearly show that PR-MPNN outperforms them by a large margin.
- We added further baselines for the heterophilic node-level datasets; see Table 11. PR-MPNNs still perform very favorably against them.
- We added detailed experiments showing that PR-MPNNs clearly improve robustness; see Table 13 and Appendix H.

If we addressed your respective changes, we kindly ask you to adjust your score. Please let us know if you have further suggestions or concerns.

---

### Author Response · Authors · 2023-11-20
**Kind reminder regarding the end of the discussion period**

Dear Reviewers,

As we approach the end of the discussion period, we thank you for your valuable feedback on our manuscript - we believe that the current revision is a significant improvement over the initial submission.

We hope our responses have effectively addressed your concerns. If so, we kindly ask you to consider revising your scores. Should any further points need clarification, we are eager to address them.

Thank you again for your time and insights!

---

### Meta-Review · Area_Chair_queP · 2023-12-05

**Metareview:**

This work provides a novel, probabilistic perspective to graph rewiring, achieving strong results both theoretically and empirically. All reviewers are in support of acceptance, and I am in agreement -- this is a great contribution to the field.

**Justification For Why Not Higher Score:**

To qualify for a spotlight, I believe the work should have compared against some more classes of graph rewiring methods (e.g. computational template-based methods such as EGP).

**Justification For Why Not Lower Score:**

All reviewers unanimously support acceptance, and I am in agreement. The results are solid.

---

### Decision · Program_Chairs · 2024-01-16

Accept (poster)